# SpatioTemporal Imputation with Graph-Informed Flow Matching

## Abstract

Missing data is a common challenge in spatiotemporal systems, arising in applications such as air quality monitoring and urban traffic management. Traditional machine learning approaches, like recurrent and graph neural networks, rely on iterative propagation, which tends to accumulate errors over time and space. Recent diffusion-based methods mitigate error propagation but require iterative sampling and often depend on problem-agnostic Gaussian priors, limiting both efficiency and effectiveness. To address these limitations, we propose **GiFlow**, a *Graph-Informed Flow Matching* framework for spatiotemporal imputation. GiFlow replaces the typical Gaussian prior with a graph-informed prior constructed via spatiotemporal filtering of observable signals, which better aligns the source distribution to the target and thereby simplifies the generation trajectory. The flow field is parameterized by a hybrid vector field model that integrates spatial attention, temporal attention, and spatiotemporal propagation, enabling joint modeling of spatial and temporal dependencies. Unlike diffusion models, GiFlow is trained via direct regression and supports deterministic, few-step generation at inference. Extensive experiments on both synthetic and real-world datasets with different missing patterns and missing rates demonstrate that the proposed GiFlow outperforms the state-of-the-art approaches in spatiotemporal imputation. The code is available in Supplementary Material.

## 1 Introduction

Spatiotemporal data characterizes both spatial and temporal information and is ubiquitous in domains such as environmental science, urban systems, and climate forecasting (Atluri et al., 2018; Wang et al., 2020). In practice, spatiotemporal data is often incomplete due to sensor failures, transmission errors, or system instability (Yi et al., 2016). The incompleteness of spatiotemporal data compromises the reliability of subsequent analyses (Ma et al., 2024; Marisca et al., 2024), motivating the need for robust spatiotemporal imputation techniques (Cao et al., 2018; Cini et al., 2022).

Early approaches to spatiotemporal imputation rely on statistical models that impose restrictive assumptions on the underlying data distribution, such as temporal smoothness levels, often failing to capture complex, nonlinear dependencies (Liu et al., 2023a; He et al., 2025). Deep learning methods have been introduced to better exploit spatiotemporal correlations. Specifically, recurrent neural networks (RNNs) are used to capture temporal dependencies by propagating hidden states (Cao et al., 2018), while graph neural networks (GNNs) are deployed to model spatial relationships over the underlying graph topology (Cini et al., 2022). Despite their success, these models generally rely on iterative propagation across space and time, which can lead to error accumulation and information bottlenecks (Deng et al., 2024; He et al., 2025; Cini et al., 2025).

Generative models provide an alternative paradigm by inferring the entire data distribution in a non-autoregressive manner. Unlike RNN/GNN-based models that propagate intermediate estimates step by step, generative models can perform imputation jointly and conditionally on all available observations, thereby avoid the accumulation of errors during iterative propagation (Liu et al., 2019; 2023a; He et al., 2025). Among them, diffusion models have demonstrated remarkable success across various domains (Croitoru et al., 2023; Yang et al., 2023; Cao et al., 2024), and recent works have adapted them for spatiotemporal imputation (Liu et al., 2023a; He et al., 2025). However, diffusion models typically rely on the problem-agnostic Gaussian prior, presenting an absence of

the available problem-specific structure. Moreover, the sampling of diffusion models requires many iterative denoising steps, and the imputation often demands multiple sampling runs followed by averaging, limiting both efficiency and robustness when applied to large-scale spatiotemporal data.

Recent work has explored flow matching (FM) as a generalization of diffusion models, which follows deterministic transport path (Lipman et al., 2023; Albergo & Vanden-Eijnden, 2023; Liu et al., 2023b). FM avoids stochastic noise injection, supports efficient deterministic sampling, and does not rely on Gaussian priors. These characteristics make FM particularly attractive for conditional tasks such as imputation, where partial observations encode strong structural information. The flexibility on prior selection allows FM to have shorter generative paths which enhances generation performance (Tong et al., 2024). Building on these insights, we propose **GiFlow**, the first *Graph-Informed Flow Matching* framework for spatiotemporal imputation. Unlike existing diffusion-based methods that rely on problem-agnostic Gaussian priors (Liu et al., 2023a; He et al., 2025), GiFlow constructs a graph-informed prior using spatiotemporal filtering of observable signals, simplifying generation trajectories. Combined with a hybrid vector field integrating attention mechanisms and spatiotemporal propagation, our approach overcomes the limitations of iterative propagation in RNN- and GNN-based models, as well as the unstructured priors and inefficiency of diffusion-based methods.

Our contributions are summarized as follows:

- We introduce GiFlow, a novel generative model for spatiotemporal imputation that integrates graph-informed priors into the flow matching framework.

- We design a graph-informed prior based on adaptive spatiotemporal filtering. Compared to the problem-agnostic Gaussian prior, this problem-tailored prior is more aligned with the target distribution and provably reduces transport cost. We also theoretically analyze the relationship between filtering factors and the receptive field in the spatiotemporal filtering process.

- We conduct extensive experiments on both synthetic and real-world datasets, demonstrating that the proposed GiFlow model achieves competitive or superior performance across diverse missing patterns and missing rates, outperforming state-of-the-art baselines.

## 2 PRELIMINARIES

### 2.1 NOTATIONS AND PROBLEM DEFINITION

**Notations.** We use calligraphic letters like $\mathcal{X}$ to represent sets, uppercase bold letters like $\mathbf{X}$ to represent matrices, and lowercase bold letters like $\mathbf{x}$ to represent vectors. We use $X_{ij}$ to represent the element in the $i$-th row and $j$-th column of $\mathbf{X}$. $\mathrm{vec}(\cdot)$ is the vectorization operation of a matrix. $\mathrm{diag}(\mathbf{x})$ represents a matrix with its diagonal elements given by vector $\mathbf{x}$. $\circ$ represents element-wise multiplication between matrices and $\oplus$ denotes the Kronecker sum operator between matrices.

**Graphs and Signals.** Let spatiotemporal data be represented as a matrix $\mathbf{X} \in \mathbb{R}^{N \times R}$, where the $r$-th column of $\mathbf{X}$ denotes the signals observed at time $r$ across $N$ nodes (*e.g.*, traffic sensors or air quality stations). We denote by $\mathcal{R} = \{1, \ldots, R\}$ the set of timesteps. The relationships among the nodes are captured by a graph $\mathcal{G} = (\mathcal{N}, \mathcal{E})$, with $\mathcal{N}$ being the set of nodes and $\mathcal{E}$ being the set of edges. Let $\mathbf{A} \in \mathbb{R}^{N \times N}$ denote the adjacency matrix, $\mathbf{D} = \mathrm{diag}(\mathbf{A1})$ the degree matrix, and $\mathbf{L} = \mathbf{D} - \mathbf{A}$ the Laplacian. For simplicity, we focus on one-dimensional signals, though the method generalizes to multi-dimensional signals.

**Spatiotemporal Imputation.** We consider scenarios where some entries of $\mathbf{X}$ are missing. Define a binary mask $\mathbf{M} \in \{0, 1\}^{N \times R}$ such that $M_{ir} = 1$ if the data on the node $i$ at time $r$ is observed, and 0 otherwise. The incomplete observations are then given by $\mathbf{X} \circ \mathbf{M}$. The task of spatiotemporal imputation is to estimate the missing entries based on the incomplete observations, leveraging both spatial dependencies across nodes and temporal dependencies across timesteps.

### 2.2 CONDITIONAL FLOW MATCHING

FM learns a vector field that transports samples from a source distribution $p_0$ to a target distribution $p_1$ (Albergo & Vanden-Eijnden, 2023; Lipman et al., 2023; Liu et al., 2023b). Let $\phi_t : [0, 1] \times \mathbb{R}^d \to \mathbb{R}^d$ denote a step-dependent flow map with $t$ being the flow step, that evolves $\mathbf{x}_0 \sim p_0$ to $\mathbf{x}_1 \sim p_1$

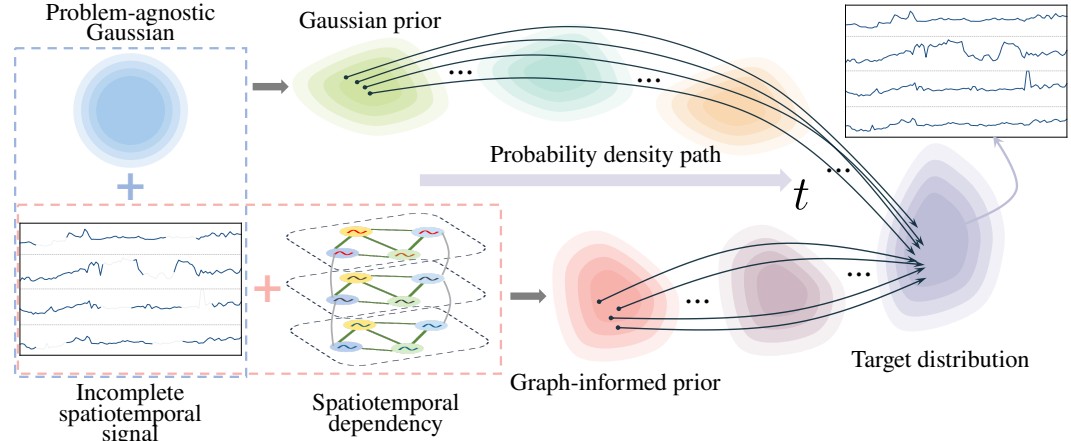

Figure 1: Schematic comparison of GiFlow and FM-Gauss (a FM model with a Gaussian prior). The red and blue dashed lines represent the information used to generate the prior for GiFlow and FM-Gauss, respectively. GiFlow constructs a graph-informed prior via adaptive spatiotemporal filtering of the observable signals, aligning the source distribution closer to the target, and hence simplifying the generation trajectories (lower part). In contrast, because a problem-agnostic Gaussian prior ignores the spatiotemporal structure, it differs significantly from the target distribution, and thereby the model must traverse a longer path to reach the target distribution (upper part).

via the ordinary differential equation (ODE):

$$d\phi_t(\mathbf{x}) = u_t(\phi_t(\mathbf{x}))dt, \quad \phi_0(\mathbf{x}) = \mathbf{x}_0, \tag{1}$$

where $u_t : [0, 1] \times \mathbb{R}^d \to \mathbb{R}^d$ is a step-dependent vector field. This induces a step-dependent probability density path $p_t$ through the push-forward operator $p_t = [\phi_t]_* p_0$ (Lipman et al., 2023).

The FM objective seeks a trainable vector field $v_t(\cdot; \boldsymbol{\theta})$ that approximates $u_t$:

$$\mathcal{L}_{\mathsf{FM}}(\boldsymbol{\theta}) = \mathbb{E}_{t \sim \mathcal{U}[0,1], \mathbf{x} \sim p_t}\big[|v_t(\mathbf{x}; \boldsymbol{\theta}) - u_t(\mathbf{x})|^2\big]. \tag{2}$$

This objective allows sampling from $p_1$ given a sample from $p_0$, as well as modeling continuous sample dynamics. However, it is generally intractable, as both $p_t$ and $u_t$ are unknown.

Conditional flow matching (CFM) (Lipman et al., 2023) provides a tractable alternative by approximating a conditional vector field $u_t(\mathbf{x} \mid \mathbf{z})$:

$$\mathcal{L}_{\mathsf{CFM}}(\boldsymbol{\theta}) = \mathbb{E}_{t \sim \mathcal{U}[0,1], \mathbf{z} \sim q(\mathbf{z}), \mathbf{x} \sim p_t(\mathbf{x}|\mathbf{z})}\big[|v_t(\mathbf{x}; \boldsymbol{\theta}) - u_t(\mathbf{x} \mid \mathbf{z})|^2\big]. \tag{3}$$

Here, $\mathbf{z}$ is chosen such that the marginal distributions of $p_t(\mathbf{x} \mid \mathbf{z})$ match the boundary distributions $p_0$ and $p_1$. Typically, $\mathbf{z} = (\mathbf{x}_0, \mathbf{x}_1)$ is sampled from a joint distribution $q(\mathbf{z}) = \pi(\mathbf{x}_0, \mathbf{x}_1)$ with marginals $p_0$ and $p_1$. Importantly, $\mathcal{L}_{\mathsf{CFM}}$ and $\mathcal{L}_{\mathsf{FM}}$ are equivalent in the sense that their gradients with respect to $\boldsymbol{\theta}$ coincide (Lipman et al., 2023).

More discussions on spatiotemporal imputation and flow matching are provided in Appendix A.

## 3 GRAPH-INFORMED FLOW MATCHING

In this section, we first construct a graph-informed prior via adaptive spatiotemporal filtering and then introduce the GiFlow framework for spatiotemporal imputation, with theoretical justifications on its effectiveness. A schematic overview of the GiFlow framework is provided in Figure 1.

### 3.1 GRAPH-INFORMED PRIOR VIA ADAPTIVE SPATIOTEMPORAL FILTERING

Let $\mathbf{X}_1^M = \mathbf{X}_1 \circ \mathbf{M}$ denote the observable spatiotemporal signal. The goal of GiFlow is to model the conditional distribution $p_1(\mathbf{X_1} \mid \mathbf{X}_1^M)$ via a generative model $p_{\boldsymbol{\theta}}(\mathbf{X_1} \mid \mathbf{X}_1^M)$. Typically, generative models start from an isotropic Gaussian and treat $\mathbf{X}_1^M$ as a conditioning variable (Liu et al., 2023a;

He et al., 2025). However, the reliance on such a simple prior complicates the generative process since the prior distribution differs significantly from the target distribution (Kollovieh et al., 2025). To construct a more structured prior, we can decompose the conditional distribution as

$$p_{\boldsymbol{\theta}}(\mathbf{X_1} \mid \mathbf{X}_1^M) = \int p_{\boldsymbol{\theta}}(\mathbf{X_1} \mid \mathbf{X}_0, \mathbf{X}_1^M) q_0(\mathbf{X}_0 \mid \mathbf{X}_1^M) \, d\mathbf{X}_0. \tag{4}$$

Setting $q_0$ to a problem-agnostic standard Gaussian, as in most existing diffusion and FM models, neglects the spatiotemporal structure. Leveraging the flexibility of FM, we construct a graph-informed prior more aligned to the target data distribution $p_1$. By aligning the source distribution to the target distribution, we aim to reduce the overall transport cost.

We construct the graph-informed prior leveraging the joint continuous spatiotemporal filtering, which has been adopted in previous joint spatiotemporal frameworks (Stanley et al., 2020; Pan et al., 2021; Einizade et al., 2024). Specifically, we consider $\mathbf{x}_1^M = \mathrm{vec}(\mathbf{X}_1^M)$ as a spatiotemporal graph signal living on top of the Cartesian product between the spatial and temporal graphs. We denote by $\mathbf{L}_\eta$ and $\mathbf{L}_\xi$ the spatial and temporal graph Laplacians, respectively. Then the spatiotemporal filtering operator can be defined as the Kronecker sum of $\mathbf{L}_\eta$ and $\mathbf{L}_\xi$, *i.e.*, $\mathbf{L}_{\eta\xi} = \tau_\xi \mathbf{L}_\xi \oplus \tau_\eta \mathbf{L}_\eta$, where $\tau_\eta$ and $\tau_\xi$ control the range of the receptive field. In this way, the joint spatiotemporal filtering operation can be described as $\mathbf{x}_{\boldsymbol{\tau}} = e^{-\mathbf{L}_{\eta\xi}} \mathbf{x}_1^M$. It can be shown (Stanley et al., 2020) that the matrix form of this spatiotemporal filtering operation, where $\mathbf{x}_{\boldsymbol{\tau}} = \mathrm{vec}(\mathbf{X}_{\boldsymbol{\tau}})$, takes the form of

$$\mathbf{X}_{\boldsymbol{\tau}} = e^{-\tau_\eta \mathbf{L}_\eta} \mathbf{X}_1^M \, e^{-\tau_\xi \mathbf{L}_\xi}. \tag{5}$$

The continuous spatiotemporal model enables adaptive filtering. From the perspective of minimizing transport cost, we obtain the optimal $\boldsymbol{\tau} = (\tau_\eta, \tau_\xi)$ by solving the following problem:

$$\underset{\tau_\eta, \tau_\xi > 0}{\text{minimize}} \ \left\| \mathbf{X}_1 - e^{-\tau_\eta \mathbf{L}_\eta} \mathbf{X}_1^M e^{-\tau_\xi \mathbf{L}_\xi} \right\|^2 + \alpha_\tau \mathrm{tr}\Big( \big( e^{-\tau_\eta \mathbf{L}_\eta} \mathbf{X}_1^M e^{-\tau_\xi \mathbf{L}_\xi} \big)^\top \mathbf{L}_\eta \, e^{-\tau_\eta \mathbf{L}_\eta} \mathbf{X}_1^M e^{-\tau_\xi \mathbf{L}_\xi} \Big), \tag{6}$$

where the first term enforces signal alignment and the second term encourages Laplacian smoothness (Bontonou et al., 2019; Dong et al., 2020). This optimization balances alignment and smoothness, producing a spatiotemporal graph-informed prior that is close to the target distribution.

Expanding the exponentials via the Taylor series gives

$$\mathbf{X}_{\boldsymbol{\tau}} = \left( \sum_{k=0}^{\infty} \frac{(-\tau_\eta)^k}{k!} \mathbf{L}_\eta^k \right) \mathbf{X}_1^M \left( \sum_{m=0}^{\infty} \frac{(-\tau_\xi)^m}{m!} \mathbf{L}_\xi^m \right), \tag{7}$$

which propagates information across all nodes and timesteps for any nonzero $(\tau_\eta, \tau_\xi)$. Truncating it to $K_\eta$ spatial hops and $K_\xi$ temporal hops gives

$$\mathbf{X}_{\boldsymbol{\tau}}^{K_\eta, K_\xi} = \left( \sum_{k=0}^{K_\eta} \frac{(-\tau_\eta)^k}{k!} \mathbf{L}_\eta^k \right) \mathbf{X}_1^M \left( \sum_{m=0}^{K_\xi} \frac{(-\tau_\xi)^m}{m!} \mathbf{L}_\xi^m \right). \tag{8}$$

**Proposition 1** (Adaptive spatiotemporal receptive field). *Let $C_s$ and $C_t$ denote the spectral radii of $\mathbf{L}_\eta$ and $\mathbf{L}_\xi$, respectively. Then the truncation error is bounded by*

$$\left\| \mathbf{X}_{\boldsymbol{\tau}} - \mathbf{X}_{\boldsymbol{\tau}}^{K_\eta, K_\xi} \right\| \leq \Bigg( \bigg( \sum_{k=K_\eta+1}^{\infty} \frac{|\tau_\eta|^k}{k!} C_s^k \bigg) \cdot \bigg( \sum_{m=0}^{\infty} \frac{|\tau_\xi|^m}{m!} C_t^m \bigg)$$
$$+ \bigg( \sum_{k=0}^{\infty} \frac{|\tau_\eta|^k}{k!} C_s^k \bigg) \cdot \bigg( \sum_{m=K_\xi+1}^{\infty} \frac{|\tau_\xi|^m}{m!} C_t^m \bigg) \Bigg) \| \mathbf{X}_1^M \|. \tag{9}$$

The proof of Proposition 1 is provided in Appendix B.1. According to Eq. (9), the truncation error can be reduced either by decreasing $(\tau_\eta, \tau_\xi)$ or increasing $(K_\eta, K_\xi)$. In particular, for smaller filtering factor $(\tau_\eta, \tau_\xi)$, a smaller truncation order $(K_\eta, K_\xi)$ suffices to achieve the same approximation error. Therefore, $(\tau_\eta, \tau_\xi)$ effectively controls the spatial and temporal receptive fields: smaller values yield more localized receptive fields, while larger values expand them to capture long-range dependencies. Optimizing $\boldsymbol{\tau} = (\tau_\eta, \tau_\xi)$ thus enables an adaptive spatiotemporal receptive field.

In the following theorem, we explicitly show how the graph-informed prior in GiFlow enables more efficient transport compared to standard FM start from an isotropic Gaussian, highlighting the benefit of incorporating structural spatiotemporal knowledge.

**Theorem 1** (Control of transport cost). *Consider flow matching for spatiotemporal imputation. Let $p_0^{\mathrm{G}}$ denote the graph-informed prior obtained via the spatiotemporal filtering operator defined in Eq. (5), with $(\tau_\eta, \tau_\xi)$ being the optimal solution to Problem (6) with $\alpha_\tau = 0$, and let $p_0^{\mathrm{Gauss}}$ be the standard isotropic Gaussian prior. Denote by $q_1$ the target distribution. Then, the transport cost of flow matching with $p_0^{\mathrm{G}}$ is no larger than that with $p_0^{\mathrm{Gauss}}$:*

$$\mathcal{C}_{\mathrm{FM}}(p_0^{\mathrm{G}} \to q_1) \leq \mathcal{C}_{\mathrm{FM}}(p_0^{\mathrm{Gauss}} \to q_1), \tag{10}$$

*where $\mathcal{C}_{\mathrm{FM}}$ denotes the expected quadratic cost along the probability path.*

The proof of Theorem 1 is provided in Appendix B.2. Intuitively, since the standard Gaussian prior ignores the spatiotemporal structure, the model must traverse a longer path to reach the target distribution. In contrast, the graph-informed prior integrates spatial smoothness and temporal consistency via adaptive spatiotemporal filtering, aligning the source distribution closer to the target distribution and thereby reducing the overall transport cost.

## 3.2 GRAPH-INFORMED PROBABILITY FLOWS

For imputation problems, the data naturally comes in pairs $(\mathbf{X}_0, \mathbf{X}_1)$ (Albergo et al., 2024). To construct an FM model, it suffices to specify a conditional probability path and a vector field. We adopt a linear conditional probability path, which is optimal in the sense that the resulting conditional flow corresponds to the optimal transport displacement map, minimizing a bound on the kinetic energy (Lipman et al., 2023). Specifically, for a data pair $(\mathbf{X}_1^M, \mathbf{X}_1)$, the graph-informed linear conditional flow is defined as

$$\phi_t(\mathbf{X} \mid \mathbf{Z}) = (1 - t)e^{-\tau_\eta \mathbf{L}_\eta} \mathbf{X}_1^M e^{-\tau_\xi \mathbf{L}_\xi} + t\mathbf{X}_1. \tag{11}$$

This induces a unique vector field:

$$u_t(\mathbf{X} \mid \mathbf{Z}) = \mathbf{X}_1 - e^{-\tau_\eta \mathbf{L}_\eta} \mathbf{X}_1^M e^{-\tau_\xi \mathbf{L}_\xi}. \tag{12}$$

Let $v_t$ be the parameterized vector field. The regression loss of GiFlow is then given by

$$\mathcal{L}_{\mathsf{GiFM}}(\boldsymbol{\theta}) = \mathbb{E}_{t \sim \mathcal{U}[0,1], \mathbf{Z} \sim q(\mathbf{Z}), \mathbf{X} \sim p_t(\mathbf{X})} \left[ \left\| \mathbf{M} \circ \left( v_t(\mathbf{X}_t; \boldsymbol{\theta}, \mathbf{M}, \mathbf{L}) - \mathbf{X}_1 + e^{-\tau_\eta \mathbf{L}_\eta} \mathbf{X}_1^M e^{-\tau_\xi \mathbf{L}_\xi} \right) \right\|^2 \right].$$

## 3.3 VECTOR FIELD MODEL

We parameterize the vector field model $v_t$ using a spatiotemporal model that captures both spatial and temporal dependencies. The architecture has three main components: spatial attention, temporal attention, and spatiotemporal propagation.

**Spatial attention.** We first learn correlations between nodes using static node embeddings. Node embeddings are processed by a GNN developed in (Morris et al., 2019) to capture spatial information. The propagated node embedding $\mathbf{X}_n$ serves as both the key and query for spatial attention. The value is computed by $\mathbf{X}_t^\eta = \mathrm{MLP}(\mathbf{X}_t) \in \mathbb{R}^{N \times H}$, where $\mathbf{X}_t$ is computed using the defined conditional flow as in Eq. (11). To learn pairwise spatial associations, we employ self-attention:

$$\mathbf{Q}^\eta = \mathbf{X}_n \mathbf{W}_Q^\eta, \quad \mathbf{K}^\eta = \mathbf{X}_n \mathbf{W}_K^\eta, \quad \mathbf{V}^\eta = \mathbf{X}_t^\eta \mathbf{W}_V^\eta, \tag{13}$$

$$\alpha_{n_1, n_2}^\eta = \frac{\exp(\langle \mathbf{q}_{n_1}^\eta, \mathbf{k}_{n_2}^\eta \rangle)}{\sum_{n' \in \mathcal{N}} \exp(\langle \mathbf{q}_{n_1}^\eta, \mathbf{k}_{n'}^\eta \rangle)}, \tag{14}$$

where $\mathbf{W}_Q^\eta, \mathbf{W}_K^\eta, \mathbf{W}_V^\eta \in \mathbb{R}^{H \times H}$ are learnable matrices, $\mathbf{Q}^\eta, \mathbf{K}^\eta, \mathbf{V}^\eta \in \mathbb{R}^{N \times H}$ denote the query, key, and value matrices for spatial self-attention, with $\mathbf{q}_i^\eta, \mathbf{k}_i^\eta, \mathbf{v}_i^\eta$ representing their $i$-th rows. For a given spatiotemporal point $(n, r)$, we aggregate spatial messages from all nodes, weighted by the learned attention scores, to obtain the spatial embedding for each node. This aggregation is computed as

$$\mathbf{h}_n^\eta = \mathrm{MLP}\left( \sum_{n' \in \mathcal{N}} \alpha_{n,n'}^\eta, \mathbf{v}_{n'}^\eta \right). \tag{15}$$

**Temporal attention.** To capture correlations across timesteps, we employ a temporal attention mechanism. Unlike recurrent sequence models such as RNNs or LSTMs, Transformers do not

inherently encode sequential information (Wen et al., 2023). To address this, we first incorporate standard positional encoding (Vaswani et al., 2017). When real-world timestamps are available, we additionally use a learnable embedding layer to encode them (Zhou et al., 2021). Let $\mathbf{X}_{PE}$ and $\mathbf{X}_{TE}$ denote the positional encoding and timestamp encoding. The input to the temporal attention module is then given by

$$\mathbf{X}_t^{\xi} = \text{MLP}(\mathbf{X}_t^{\top}) + \mathbf{X}_{PE} + \mathbf{X}_{TE} \in \mathbb{R}^{R \times H}. \tag{16}$$

For any pair of timesteps $(r_1, r_2) \in \mathcal{R}$, $\mathbf{X}_t^{\xi}$ serves as the key, query, and value in the temporal attention computation. The temporal attention scores and aggregation are computed as

$$\mathbf{Q}^{\xi} = \mathbf{X}_t^{\xi}\mathbf{W}_Q^{\xi}, \quad \mathbf{K}^{\xi} = \mathbf{X}_t^{\xi}\mathbf{W}_K^{\xi}, \quad \mathbf{V}^{\xi} = \mathbf{X}_t^{\xi}\mathbf{W}_V^{\xi}, \tag{17}$$

$$\alpha_{r_1,r_2}^{\xi} = \frac{\exp\left(\langle \mathbf{q}_{r_1}^{\xi}, \mathbf{k}_{r_2}^{\xi} \rangle\right)}{\sum_{r' \in \mathcal{R}} \exp\left(\langle \mathbf{q}_{r_1}^{\xi}, \mathbf{k}_{r'}^{\xi} \rangle\right)}, \tag{18}$$

where $\mathbf{W}_Q^{\xi}, \mathbf{W}_K^{\xi}, \mathbf{W}_V^{\xi} \in \mathbb{R}^{H \times H}$ are learnable parameter matrices, and $\mathbf{Q}^{\xi}, \mathbf{K}^{\xi}, \mathbf{V}^{\xi} \in \mathbb{R}^{R \times H}$ denote the query, key, and value matrices. For a spatiotemporal point $(n, r)$, temporal messages from all timesteps are aggregated using the learned attention weights, producing the temporal embedding:

$$\mathbf{h}_r^{\xi} = \text{MLP}\left(\sum_{r' \in \mathcal{R}} \alpha_{r,r'}^{\xi}, \mathbf{v}_{r'}^{\xi}\right). \tag{19}$$

**Spatiotemporal propagation.** The aggregated spatial and temporal messages are concatenated with the original features and time embedding that encodes the information of the step $t$ in the probability density path, then projected via a linear layer to obtain $\mathbf{H} \in \mathbb{R}^{N \times R \times H}$. We then perform $L_{MP}$ layers of message passing in both spatial and temporal domains, with the $\ell$-th ($\ell = 1, \ldots, L_{MP}$) layer defined by

$$\begin{aligned}\mathbf{H}_r^{(\ell+1)} &= \text{GNN}(\mathbf{H}_r, \mathcal{G}_s) \in \mathbb{R}^{N \times H}, \quad \forall r \in \mathcal{R}, \\ \mathbf{H}_n^{(\ell+1)} &= \text{GNN}(\mathbf{H}_n, \mathcal{G}_t) \in \mathbb{R}^{R \times H}, \quad \forall n \in \mathcal{N},\end{aligned} \tag{20}$$

where spatial message passing is applied independently for each timestep, and temporal message passing is applied independently for each node. The GNN model follows the architecture developed in (Wu et al., 2019). After $L_{MP}$ layers, we obtain $\mathbf{H}^{prop} \in \mathbb{R}^{N \times R \times H}$. Then a linear layer projects the features back to the original signal dimension. For one-dimensional signals, the final output is

$$\mathbf{X}^{out} = \text{MLP}(\mathbf{H}^{prop}) \in \mathbb{R}^{N \times R}. \tag{21}$$

The GiFlow model integrates graph-informed priors with a spatiotemporal architecture in the flow matching framework, providing an effective generative model for spatiotemporal imputation.

## 4 EXPERIMENTS

We assess the performance of GiFlow using synthetic data (Qiu et al., 2017; Giraldo et al., 2022) as well as four widely used real-world datasets that have different sizes and spatiotemporal patterns: two air quality datasets (Air-36 and AQI) (Zheng et al., 2015; Yi et al., 2016) and two traffic datasets (PeMS04 and PeMS08) (Guo et al., 2021). Specifically, Air-36 and AQI collect hourly sampled PM2.5 pollutant data in China. PeMS04 and PeMS08 are collected by the Caltrans Performance Measurement System (PeMS) (Chen et al., 2001), containing highway traffic flow data in California. Both datasets originally collect data every 30 seconds, and the collected data is then aggregated with a 5-minute interval. To simulate realistic incomplete spatiotemporal signals, we adopt two missing data injection strategies: (1) Point missing: following the setup of (Cini et al., 2022; Deng et al., 2024), we randomly mask a fraction $\rho$ of the available data; (2) Block missing: we first randomly select a node and a starting timestep, then mask a contiguous segment of data from that timestep for the selected node. This process is repeated iteratively until a fraction $\rho$ of the available data is masked. We compare the performance of GiFlow with five non-parametric methods (Mean-S, Mean-T, Linear, KNN, and FP (Rossi et al., 2022)), two RNN-based methods (BRITS (Cao et al., 2018) and SAITS (Du et al., 2023)), four spatiotemporal GNN-based and transformer-based methods (SPIN (Marisca et al., 2022), GRIN (Cini et al., 2022), OPCR (Deng et al., 2024), and

Table 1: Imputation performance with point missing strategy ($\rho = 20\%$).

| Model | Air-36 | | | AQI | | |
|---|---|---|---|---|---|---|
| | MAE | RMSE | MAPE | MAE | RMSE | MAPE |
| Mean-S | $19.22^{***}_{\pm0.17}$ | $31.81^{***}_{\pm0.50}$ | $45.60^{***}_{\pm0.75}$ | $34.93^{***}_{\pm0.05}$ | $48.94^{***}_{\pm0.53}$ | $112.45^{***}_{\pm0.27}$ |
| Mean-T | $30.39^{***}_{\pm0.22}$ | $44.83^{***}_{\pm0.55}$ | $83.64^{***}_{\pm1.22}$ | $20.57^{***}_{\pm0.02}$ | $33.78^{***}_{\pm0.31}$ | $55.44^{***}_{\pm0.18}$ |
| Linear | $11.02^{***}_{\pm0.12}$ | $21.28^{***}_{\pm0.60}$ | $27.68^{***}_{\pm0.50}$ | $8.97^{***}_{\pm0.04}$ | $19.95^{***}_{\pm0.25}$ | $21.42^{***}_{\pm0.11}$ |
| KNN | $20.33^{***}_{\pm0.14}$ | $34.25^{***}_{\pm0.71}$ | $41.74^{***}_{\pm0.96}$ | $18.95^{***}_{\pm0.03}$ | $33.03^{***}_{\pm0.27}$ | $52.44^{***}_{\pm0.20}$ |
| FP | $16.51^{***}_{\pm0.17}$ | $28.68^{***}_{\pm0.80}$ | $38.51^{***}_{\pm1.70}$ | $15.65^{***}_{\pm0.02}$ | $27.20^{***}_{\pm0.41}$ | $45.70^{***}_{\pm0.18}$ |
| BRITS | $14.23^{***}_{\pm0.17}$ | $24.64^{***}_{\pm0.59}$ | $31.42^{***}_{\pm0.72}$ | $16.55^{***}_{\pm0.12}$ | $26.78^{***}_{\pm0.36}$ | $41.25^{***}_{\pm0.35}$ |
| SAITS | $14.32^{***}_{\pm0.13}$ | $23.85^{***}_{\pm0.61}$ | $31.62^{***}_{\pm0.85}$ | $17.95^{***}_{\pm0.07}$ | $28.95^{***}_{\pm0.41}$ | $44.89^{***}_{\pm0.30}$ |
| SPIN | $11.05^{*}_{\pm0.87}$ | $20.97^{*}_{\pm1.90}$ | $22.26_{\pm1.56}$ | $8.72^{***}_{\pm0.12}$ | $19.61^{*}_{\pm0.67}$ | $18.55^{***}_{\pm0.28}$ |
| GRIN | $9.94^{*}_{\pm0.12}$ | $19.09_{\pm0.87}$ | $21.95^{*}_{\pm0.22}$ | $7.97^{*}_{\pm0.08}$ | $18.46_{\pm0.54}$ | $16.81^{*}_{\pm0.24}$ |
| OPCR | $\underline{10.03}^{*}_{\pm0.12}$ | $19.32^{*}_{\pm0.60}$ | $21.61_{\pm0.68}$ | $8.40^{*}_{\pm0.20}$ | $19.30^{*}_{\pm0.69}$ | $16.91_{\pm0.59}$ |
| PriSTI | $10.29^{***}_{\pm0.14}$ | $19.66^{*}_{\pm0.25}$ | $\underline{21.91}^{*}_{\pm0.66}$ | $8.17_{\pm0.28}$ | $19.85^{***}_{\pm0.28}$ | $\underline{16.37}_{\pm0.59}$ |
| ImputeFormer | $10.18_{\pm0.73}$ | $19.55_{\pm1.26}$ | $22.74_{\pm2.52}$ | $\underline{7.90}_{\pm0.33}$ | $17.96_{\pm0.36}$ | $16.47_{\pm0.89}$ |
| CoFILL | $10.03_{\pm0.45}$ | $19.74^{*}_{\pm0.93}$ | $23.32^{***}_{\pm0.47}$ | OOT | OOT | OOT |
| GiFlow | $\mathbf{9.54}_{\pm0.18}$ | $\mathbf{18.10}_{\pm0.78}$ | $\mathbf{21.27}_{\pm0.33}$ | $\mathbf{7.83}_{\pm0.10}$ | $\mathbf{17.80}_{\pm0.28}$ | $\mathbf{16.24}_{\pm0.31}$ |

$*$ represents a $p$-value lower than 0.05 (significant difference from GiFlow)

$***$ represents a $p$-value lower than 0.001 (highly significant difference from GiFlow)

ImputeFormer (Nie et al., 2024)), and two diffusion-based method (PriSTI (Liu et al., 2023a) and CoFILL (He et al., 2025)). To obtain the filtering factors $\tau_\eta$ and $\tau_\xi$, we optimize Problem (6) with stochastic gradient descent. Specifically, the filtering factors are optimized using the training data, where the complete ground-truth signals are available. Once selected, the filtering factors remain constant during inference. Details about the datasets and baselines are provided in Appendix C.1 and Appendix C.2, respectively. To evaluate the performance, we use three metrics: mean absolute error (MAE), root mean squared error (RMSE), and mean absolute percentage error (MAPE). All the experiment results are conducted five times using different seeds, and we report the average performance. The implementation details can be found in Appendix C.3.

## 4.1 Performance Evaluation on Synthetic Datasets

To evaluate the proposed GiFlow, we generate a synthetic spatiotemporal dataset following the procedure described in (Qiu et al., 2017; Giraldo et al., 2022). This yields a smooth, temporally evolving graph signal $\mathbf{X} \in \mathbb{R}^{N \times R}$. Specifically, we sample 50 nodes uniformly at random within a 50×50 square domain. A graph is then constructed using KNN based on the spatial positions, with $k = 5$. Let $\mathbf{L}_\eta \in \mathbb{R}^{N \times N}$ be the spatial graph Laplacian, for which we compute the eigen-decomposition $\mathbf{L}_\eta = \mathbf{V}\mathbf{\Lambda}\mathbf{V}^\top$. Its inverse square root is used to construct a smoothed propagation operator $\mathbf{L}_\eta^{-\frac{1}{2}} = \mathbf{U}\mathbf{\Lambda}^{-\frac{1}{2}}\mathbf{U}^\top$ where $\mathbf{\Lambda}^{-\frac{1}{2}} = \mathrm{diag}(0, \lambda_2^{-\frac{1}{2}}, \ldots, \lambda_N^{-\frac{1}{2}})$. The initial signal is generated in the spectral domain as a low-frequency signal. Subsequent signals are generated iteratively via $\mathbf{x}_r = \mathbf{x}_{r-1} + \mathbf{L}_\eta^{-\frac{1}{2}}\mathbf{f}_r$, where $\mathbf{f}$ is an i.i.d. Gaussian signal. The length of the generated signal is set to $R = 3000$. To simulate the real-world noisy conditions, we add small Gaussian noise to the signal using $\tilde{\mathbf{X}} = \mathbf{X} + \boldsymbol{\epsilon}$, $\boldsymbol{\epsilon} \sim \mathcal{N}(\mathbf{0}, \sigma^2\mathbf{I})$. We conduct experiments using the point missing strategy ($\rho = 20\%$) with both $\sigma = 0.1$ and $\sigma = 0.3$ to evaluate the performance under different noisy levels. The results are given in Table 2, where the best results are in **bold**, and the second best results are underlined. From the results, we can see that the GiFlow model performs well under different noisy levels.

Table 2: Imputation performance on the synthetic dataset.

| Model | $\sigma = 0.1$ | | | $\sigma = 0.3$ | | |
|---|---|---|---|---|---|---|
| | MAE | RMSE | MAPE | MAE | RMSE | MAPE |
| BRITS | 0.35 | 0.56 | 7.90 | 0.39 | 0.64 | 12.26 |
| SAITS | 0.30 | 0.41 | 7.52 | 0.36 | 0.47 | 11.27 |
| SPIN | 0.83 | 1.08 | 26.83 | 0.87 | 1.12 | 29.82 |
| GRIN | $\underline{0.24}$ | $\underline{0.31}$ | $\mathbf{5.98}$ | $\underline{0.35}$ | $\underline{0.46}$ | $\underline{11.05}$ |
| OPCR | 0.32 | 0.42 | 11.12 | 0.44 | 0.57 | 14.88 |
| PriSTI | 0.32 | 0.36 | 11.21 | 0.37 | 0.47 | 12.48 |
| GiFlow | $\mathbf{0.23}$ | $\mathbf{0.30}$ | $\underline{6.65}$ | $\mathbf{0.34}$ | $\mathbf{0.44}$ | $\mathbf{10.67}$ |

Table 3: Imputation performance with block missing strategy ($\rho = 20\%$).

| Model | Air-36 | | | AQI | | |
|---|---|---|---|---|---|---|
| | MAE | RMSE | MAPE | MAE | RMSE | MAPE |
| Mean-S | $19.80^{***}_{\pm0.29}$ | $32.03^{*}_{\pm0.23}$ | $44.00^{***}_{\pm1.88}$ | $34.96^{***}_{\pm0.19}$ | $48.70^{***}_{\pm0.91}$ | $117.44^{***}_{\pm1.95}$ |
| Mean-T | $41.06^{***}_{\pm0.42}$ | $59.70^{***}_{\pm1.00}$ | $102.80^{***}_{\pm2.19}$ | $26.81^{***}_{\pm0.13}$ | $42.69^{***}_{\pm0.38}$ | $76.71^{***}_{\pm1.19}$ |
| Linear | $33.03^{***}_{\pm0.57}$ | $52.53^{***}_{\pm1.28}$ | $81.95^{***}_{\pm5.11}$ | $23.87^{***}_{\pm0.24}$ | $40.83^{***}_{\pm0.75}$ | $69.03^{***}_{\pm1.12}$ |
| KNN | $20.78^{***}_{\pm0.39}$ | $34.33^{***}_{\pm0.56}$ | $40.34^{***}_{\pm2.57}$ | $18.58^{***}_{\pm0.12}$ | $32.80^{***}_{\pm0.36}$ | $52.55^{***}_{\pm1.13}$ |
| FP | $16.65^{***}_{\pm0.30}$ | $28.66^{*}_{\pm0.74}$ | $35.76^{***}_{\pm2.02}$ | $15.19^{***}_{\pm0.14}$ | $26.74_{\pm0.53}$ | $44.81^{***}_{\pm1.08}$ |
| BRITS | $17.78^{***}_{\pm0.46}$ | $28.16^{*}_{\pm0.93}$ | $32.19^{*}_{\pm2.23}$ | $17.60^{***}_{\pm0.09}$ | $28.24_{\pm0.19}$ | $44.21^{***}_{\pm0.21}$ |
| SAITS | $18.04^{***}_{\pm0.23}$ | $28.63^{*}_{\pm0.52}$ | $40.54^{***}_{\pm1.01}$ | $19.45^{***}_{\pm0.16}$ | $30.85^{*}_{\pm0.66}$ | $49.86^{***}_{\pm0.79}$ |
| SPIN | $16.59^{***}_{\pm0.31}$ | $28.07^{*}_{\pm0.37}$ | $31.17^{*}_{\pm1.04}$ | $14.73^{***}_{\pm0.15}$ | $26.79_{\pm0.47}$ | $32.23_{\pm0.21}$ |
| GRIN | $16.27^{***}_{\pm0.32}$ | $27.67^{*}_{\pm0.67}$ | $31.86^{*}_{\pm1.17}$ | $14.47^{***}_{\pm0.14}$ | $\mathbf{25.23^{***}_{\pm0.23}}$ | $33.03_{\pm0.32}$ |
| OPCR | $15.27^{*}_{\pm0.19}$ | $25.44^{*}_{\pm1.10}$ | $33.38^{***}_{\pm1.03}$ | $14.52^{*}_{\pm0.28}$ | $25.95_{\pm1.52}$ | $31.75_{\pm0.54}$ |
| PriSTI | $\underline{15.07}_{\pm0.65}$ | $\underline{25.57}_{\pm1.27}$ | $29.84_{\pm1.96}$ | $14.54^{*}_{\pm0.45}$ | $26.61_{\pm1.48}$ | $31.77_{\pm0.39}$ |
| ImputeFormer | $15.41_{\pm0.63}$ | $27.88_{\pm1.61}$ | $\underline{29.46}_{\pm4.38}$ | $\underline{14.02}_{\pm0.46}$ | $25.68_{\pm0.86}$ | $\underline{31.58}_{\pm1.58}$ |
| GiFlow | $\mathbf{14.76}_{\pm0.38}$ | $\mathbf{25.33}_{\pm2.14}$ | $\mathbf{28.95}_{\pm0.80}$ | $\mathbf{13.74}_{\pm0.30}$ | $\underline{25.43}_{\pm2.61}$ | $\mathbf{31.09}_{\pm1.79}$ |

$^{*}$ represents a $p$-value lower than 0.05 (significant difference from GiFlow)
$^{***}$ represents a $p$-value lower than 0.001 (highly significant difference from GiFlow)

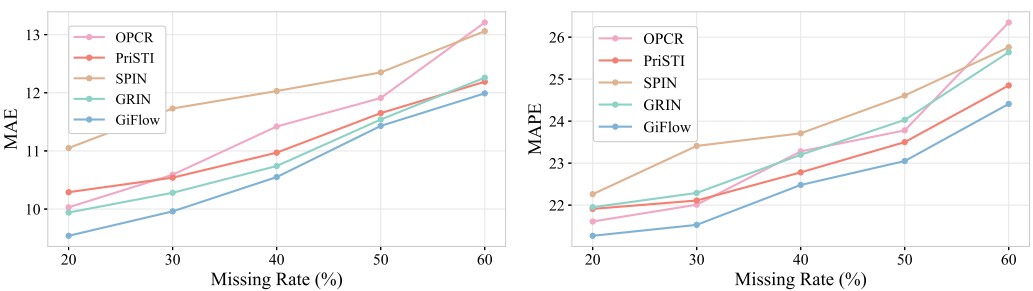

Figure 2: Performance with different missing rates.

## 4.2 PERFORMANCE EVALUATION ON REAL-WORLD DATASETS

We first evaluate model performance on the two air quality datasets: a small Air-36 dataset that is collected in Beijing and a large dataset that is collected from 43 Chinese cities. The results with the point missing and block missing strategies ($\rho = 20\%$) are reported in Table 1 and 3, respectively. OOT represents models that do not finish the training within 24 hours. We run $t$-tests comparing GiFlow with each baseline method to verify the statistical significance of our results. Stars show when a baseline performs significantly different than GiFlow ($^{*}$ for $p$-value lower than 0.05 and $^{***}$ for $p$-value lower than 0.001). Most baselines show significant differences on both datasets, confirming that our gains are meaningful. For example, all methods except ImputeFormer and CoFILL show significantly lower performance on Air-36 in Table 1. The results show that simple averaging methods, Mean-S and Mean-T, fail to achieve satisfactory results, indicating that simple spatial or temporal averaging cannot capture the system dynamics. Linear interpolation performs well under point missing, but degrades significantly under block missing. The same happens to RNN-based methods, *i.e.*, BRITS and SAITS, which even underperform the nonparametric FP model. This is because for block missing, there are contiguous missing blocks, making methods that only rely on individual time series struggle with inferring missing values based on signals from distant timesteps. The other deep learning methods that consider both spatial and temporal information, *i.e.*, SPIN, GRIN, OPCR, PriSTI, and GiFlow, perform better than the methods using only temporal information. Among them, GiFlow achieves the overall best results across different missing patterns and metrics, demonstrating its effectiveness in spatiotemporal imputation.

To further evaluate the effectiveness of the proposed GiFlow model, we conduct experiments on the Air-36 dataset, using the point missing strategy with $\rho$ ranging from 20% to 60%. For comparison, we choose four spatiotemporal baselines, which are shown to be better than other baselines

Table 4: Imputation performance with point missing strategy on traffic data.

| Model | PeMS04 | | | PeMS08 | | |
|---|---|---|---|---|---|---|
| | MAE | RMSE | MAPE | MAE | RMSE | MAPE |
| Mean-S | $89.81^{***}_{\pm 0.08}$ | $117.55^{***}_{\pm 0.10}$ | $83.53^{***}_{\pm 0.39}$ | $86.64^{***}_{\pm 0.11}$ | $113.59^{***}_{\pm 0.12}$ | $141.40^{***}_{\pm 1.67}$ |
| Mean-T | $26.23^{***}_{\pm 0.02}$ | $39.74^{***}_{\pm 0.07}$ | $20.02^{***}_{\pm 0.15}$ | $21.27^{***}_{\pm 0.05}$ | $39.74^{***}_{\pm 0.07}$ | $13.96^{***}_{\pm 0.06}$ |
| Linear | $18.29^{***}_{\pm 0.02}$ | $29.87^{***}_{\pm 0.08}$ | $12.82^{***}_{\pm 0.08}$ | $14.71^{***}_{\pm 0.04}$ | $24.04^{***}_{\pm 0.08}$ | $9.94^{***}_{\pm 0.05}$ |
| KNN | $98.28^{***}_{\pm 0.09}$ | $131.83^{***}_{\pm 0.17}$ | $95.47^{***}_{\pm 0.40}$ | $117.65^{***}_{\pm 0.19}$ | $152.50^{***}_{\pm 0.22}$ | $195.18^{***}_{\pm 2.10}$ |
| FP | $120.53^{***}_{\pm 0.11}$ | $155.00^{***}_{\pm 0.08}$ | $167.39^{***}_{\pm 0.78}$ | $118.93^{***}_{\pm 0.08}$ | $149.14^{***}_{\pm 0.30}$ | $206.13^{***}_{\pm 2.96}$ |
| BRITS | $22.14^{***}_{\pm 0.09}$ | $37.33^{***}_{\pm 0.16}$ | $17.08^{***}_{\pm 0.22}$ | $17.66^{***}_{\pm 0.03}$ | $29.13^{***}_{\pm 0.08}$ | $12.07^{***}_{\pm 0.10}$ |
| SAITS | $24.85^{***}_{\pm 0.29}$ | $40.39^{***}_{\pm 0.45}$ | $18.03^{***}_{\pm 0.40}$ | $18.15^{***}_{\pm 0.13}$ | $28.28^{***}_{\pm 0.17}$ | $12.48^{***}_{\pm 0.27}$ |
| SPIN | $18.21^{***}_{\pm 0.07}$ | $30.49^{***}_{\pm 0.08}$ | $12.34^{***}_{\pm 0.24}$ | $14.82^{***}_{\pm 0.06}$ | $24.26^{***}_{\pm 0.43}$ | $9.34^{***}_{\pm 0.06}$ |
| GRIN | $\mathbf{16.28}_{\pm \mathbf{0.13}}$ | $26.79_{\pm 0.33}$ | $\mathbf{11.12}_{\pm \mathbf{0.32}}$ | $13.72^{***}_{\pm 0.12}$ | $21.49^{***}_{\pm 0.29}$ | $8.82^{*}_{\pm 0.27}$ |
| OPCR | $16.29_{\pm 0.05}$ | $\mathbf{26.55}_{\pm \mathbf{0.05}}$ | $11.16_{\pm 0.15}$ | $12.77_{\pm 0.20}$ | $19.88_{\pm 0.32}$ | $8.67_{\pm 0.52}$ |
| PriSTI | $16.66_{\pm 0.09}$ | $27.15^{*}_{\pm 0.05}$ | $11.82^{*}_{\pm 0.32}$ | $13.02_{\pm 0.37}$ | $20.08_{\pm 0.45}$ | $8.79_{\pm 0.86}$ |
| GiFlow | $16.39_{\pm 0.27}$ | $26.76_{\pm 0.31}$ | $11.15_{\pm 0.24}$ | $\mathbf{12.66}_{\pm \mathbf{0.19}}$ | $\mathbf{19.83}_{\pm \mathbf{0.05}}$ | $\mathbf{8.43}_{\pm \mathbf{0.18}}$ |

* represents a $p$-value lower than 0.05 (significant difference from GiFlow)
*** represents a $p$-value lower than 0.001 (highly significant difference from GiFlow)

according to the results reported in Table 1 and 3. The results on MAE and MAPE are presented in Figure 2, while the results on RMSE are presented in Appendix C.4. From the results, we observe that for all the methods, the imputation performance steadily degrades with increasing missing rates. Moreover, the proposed GiFlow model consistently outperforms the other methods across different missing rates. These results validate the robustness of the proposed GiFlow model under different missing patterns and missing rates.

As proved in Proposition 1, the filtering factors $\tau_\eta$ and $\tau_\xi$ determine the receptive field of the spatiotemporal filtering operation. Intuitively, when facing a higher missing rate, the model would require a larger receptive field to obtain a close approximation of the missing signals based on the observable ones. In the following, we investigate how $\tau_\eta$ and $\tau_\xi$ change as the missing rate $\rho$ increases. The values of the filtering factors $\tau_\eta$ and $\tau_\xi$ under both the point missing strategy and block missing strategy on the Air-36 dataset with $\rho$ ranging from 20% to 60% are presented in Figure 3. It is evident that with increasing missing rate, $\tau_\eta$ and $\tau_\xi$ become

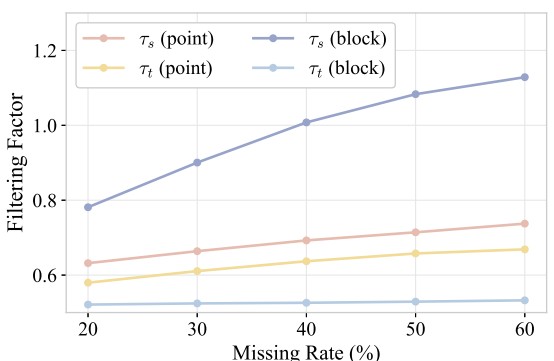

Figure 3: Filtering factors with different missing rate.

larger and larger. This pattern corroborates the results we obtained from Proposition 1 that with increasing missing rate, the model likely requires a larger spatiotemporal receptive field. Moreover, for block missing, $\tau_\eta$ increases a lot as the missing rate increases, while $\tau_\xi$ remains relatively stable. This is largely due to the fact that with the block missing strategy, we have larger temporal gaps in the data, making the model more reliant on spatial filtering. A similar phenomenon has been observed in the AQI dataset, which is presented in Appendix C.5.

### 4.3 ABLATION STUDY

In the proposed GiFlow model, the graph-informed prior construction is critical as it provides a close alignment between the source and target distribution, and hence reduces the transport cost. To empirically validate the effectiveness of the graph-informed prior generated with adaptive spatiotemporal filtering, we evaluate several variants of GiFlow. Specifically, we consider: (1) FM-Gauss: a FM model employs the same vector field model architecture as in GiFlow but with a problem-agnostic Gaussian prior; (2) GFM: GiFlow with a spatial-only graph-informed prior, *i.e.*, the filtering parameter $\tau_\eta$ is obtained by optimizing Problem (6) with $\tau_\xi$ fixed as 0; (3) TFM: GiFlow with a temporal-

only graph-informed prior, *i.e.*, the filtering parameter $\tau_\xi$ is obtained by optimizing Problem (6) with $\tau_\eta$ fixed as 0. We conduct experiments on Air-36 dataset with point missing strategy. To validate the results in Theorem 1, we also evaluate the transport cost of different models. The results are reported in Table 5. From the results, we observe that the FM-Gauss performs the worst, and it is worse than several baselines in Table 1, emphasizing that FM with a Gaussian prior fails to achieve state-of-the-art spatiotemporal imputation performance. TFM and GFM give better results than FM-Gauss, indicating that the Laplacian filtering in both the spatial and temporal domains provides more structured and informative prior to the FM framework. Notably, GFM outperforms all the baselines in Table 1, indicating that the spatial filtering alone already provides a good prior for the model. Leveraging on both spatial and temporal dependencies, GiFlow gives the best results, indicating that combining both spatial and temporal filtering brings additional performance gain. The results on transport cost corroborate the conclusions of Theorem 1. They demonstrate that applying graph filtering can substantially reduce the transport cost. Moreover, it can also be observed that models with lower transport cost tend to achieve better performance.

Table 5: The effect of different priors. (T. C. stands for transport cost.)

| Model | T. C. | MAE | RMSE | MAPE |
|---|---|---|---|---|
| FM-Gauss | 299.62 | $12.79^{***}_{\pm 0.63}$ | $22.15^{***}_{\pm 1.18}$ | $26.85^{***}_{\pm 1.92}$ |
| TFM | 123.39 | $10.12^{***}_{\pm 0.13}$ | $19.60^{***}_{\pm 0.72}$ | $22.41^{***}_{\pm 0.82}$ |
| GFM | 115.05 | $9.75^{*}_{\pm 0.23}$ | $18.67_{\pm 0.72}$ | $21.55^{*}_{\pm 0.54}$ |
| GiFlow | 104.29 | $\mathbf{9.54}_{\pm 0.18}$ | $\mathbf{18.10}_{\pm 0.78}$ | $\mathbf{21.27}_{\pm 0.33}$ |

### 4.4 APPLICABILITY TO OTHER DATASETS

In this section, we evaluate GiFlow on two traffic datasets about highway traffic flow in California to validate its applicability to other datasets. Specifically, we conduct experiments using the point missing strategy with $\rho = 20\%$ on the PeMS04 and PeMS08 datasets. The results are reported in Table 4. It can be observed that KNN and FP perform quite bad, indicating that relying only on the spatial dependencies cannot characterize the system dynamics well. The spatiotemporal methods still achieve good results, highlighting the importance of considering both spatial and temporal dependencies. Among the spatiotemporal approaches, GiFlow achieves superior or competitive results compared to other methods, validating the applicability of GiFlow to other datasets. On PeMS04, although GiFlow does not achieve the best result, the best-performed model shows no significant difference from GiFlow, indicating competitive but not always superior performance. More results on traffic datasets with block missing strategy are provided in Appendix 8.

### 4.5 RUNTIME ANALYSIS

In this section, we evaluate the complexity of different generative models. Specifically, we evaluate the inference time (in minutes) required to impute the test sets of the datasets for the considered diffusion models and GiFlow. The experiments are conducted on an A100 NVIDIA GPU with 80GB of memory. The results are reported in Table 6. From the results, it is evident that GiFlow significantly outperforms diffusion models in terms of inference speed.

Table 6: The inference time on test set.

| Model | Air-36 | AQI | PeMS04 | PeMS08 |
|---|---|---|---|---|
| PriSTI | 9.30 | 43.12 | 15.58 | 7.46 |
| CoFILL | 167.44 | 384.08 | 485.91 | 282.06 |
| GiFlow | 0.28 | 2.47 | 0.59 | 0.99 |

## 5 CONCLUSION

In this work, we consider the problem of spatiotemporal imputation. We developed a graph-informed flow matching method named GiFlow, which uses a graph-informed prior derived based on adaptive spatiotemporal filtering of observable signals. Compared with the problem-agnostic Gaussian prior, the proposed graph-informed prior better aligns with the target distribution, and it provably reduces the transport cost from the source to the target distribution. We also theoretically analyze the relationship between spatiotemporal filtering factors and the receptive field in the filtering process. Experiments on both synthetic and real-world datasets with various missing patterns and missing rates demonstrate the effectiveness and the robustness of the proposed GiFlow model.

## REPRODUCIBILITY STATEMENT

For the developed theoretical results, we have clearly mentioned the assumptions, and complete proofs are given in Appendix B. For the experiments, we use open-sourced data and we provide a detailed description in Appendix C.1. For the implementation, we provide implementation details in C.3 and the code is available in Supplementary Material.

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

# A RELATED WORK

Spatiotemporal imputation addresses the problem of reconstructing missing values in data that combine temporal dynamics with spatial dependencies, with applications in domains such as air quality monitoring (Cao et al., 2018), traffic forecasting (Li et al., 2018), and weather prediction (Price et al., 2025). Early statistical approaches rely on distributional assumptions such as temporal smoothness or local similarity across series. Examples include autoregressive models (Ansley & Kohn, 1984), expectation-maximization (Nelwamondo et al., 2007), and $k$-nearest neighbors (Beretta & Santaniello, 2016). While these methods are simple and theoretically well-understood, they struggle in real-world settings where spatial and temporal dependencies are highly nonlinear and heterogeneous. Their limited capacity to capture complex interactions has motivated the development of more flexible machine learning approaches.

Neural methods have significantly advanced imputation by better capturing temporal and spatial dependencies. RNNs and their extensions exploit temporal correlations by recursively propagating hidden states, as in BRITS (bidirectional recurrent imputation for time series) (Cao et al., 2018), while transformer-based designs such as SAITS (self-attention-based imputation for time series) (Du et al., 2023) introduce attention mechanisms to jointly optimize imputation and reconstruction. However, these approaches typically operate on individual time series and ignore spatial relationships. To address this limitation, GNN models have been proposed to incorporate spatial dependencies by propagating information over a graph topology. Examples include GRIN (graph recurrent imputation network) (Cini et al., 2022), SPIN (spatiotemporal point inference network) (Marisca et al., 2022), and OPCR (one-step propagation and confidence-based refinement) (Deng et al., 2024), which combine temporal modeling with graph-based message passing. Despite their effectiveness, these iterative propagation schemes are prone to error accumulation and information bottlenecks, particularly under high missing rates (Cini et al., 2025).

Generative models provide an alternative paradigm by directly modeling conditional data distributions, thereby avoiding the accumulation of errors across propagation steps. Diffusion-based approaches (Sohl-Dickstein et al., 2015; Ho et al., 2020; Song et al., 2021) have demonstrated strong generative performance in multiple domains, from vision (Dhariwal & Nichol, 2021; Rombach et al., 2022) to spatiotemporal data (Liu et al., 2023a; He et al., 2025). Conditional diffusion frameworks, such as CSDI (conditional score-based diffusion models for imputation) (Tashiro et al., 2021), have been adapted to time-series imputation, while PriSTI (spatiotemporal imputation with enhanced prior modeling) (Liu et al., 2023a) and related models extend them to spatiotemporal settings. However, these methods typically rely on problem-agnostic Gaussian priors and require computationally expensive iterative denoising. In practice, accurate inference often demands multiple sampling runs followed by averaging, which limits both efficiency and robustness when applied to large-scale spatiotemporal data.

The iterative nature of diffusion models leads to high inference computational costs. To improve the sampling efficiency of diffusion models, consistency models have been proposed, where the models are trained to map any point at any time step to the trajectory's starting point (Song et al., 2023). The idea of the consistency model has also been applied to the problem of spatiotemporal imputation, which demonstrates significant improvement in inference computational costs (Solís-García et al., 2025). This line of approaches, however, still relies on the problem-agnostic Gaussian priors. It has been shown that the idea of the consistency model can be extended to accommodate the flow matching framework as well (Liu et al., 2025b). But the effectiveness of such an extension has not been investigated yet in the spatiotemporal imputation task.

Flow matching (Albergo & Vanden-Eijnden, 2023; Lipman et al., 2023; Liu et al., 2023b) generalizes diffusion by directly learning a continuous probability flow from a source to a target distribution, regressing vector fields along transport paths. Instead of relying on stochastic noise injection, flow matching directly learns a continuous probability flow that transports a source distribution to the target distribution. FM can accommodate arbitrary source distributions, although Gaussian priors are still often chosen for convenience. FM avoids stochastic noise injection, reduces training variance, and stabilizes optimization (Lipman et al., 2023; Albergo & Vanden-Eijnden, 2023). Moreover, the deterministic inference of FM enables efficient sampling without repeated averaging as required in diffusion models (Liu et al., 2023a). These characteristics make FM particularly suitable for tasks where partial observations are available (Albergo et al., 2024; Kollovieh et al., 2025; Liu et al.,

## B PROOFS

### B.1 PROOF OF PROPOSITION 1

Our results on the truncation error analysis of the spatiotemporal filtering can be viewed as an extension of the truncation error analysis for spatial filtering presented in (Behmanesh et al., 2023).

The Taylor series for the spatiotemporal filtering defined in (5) is given by

$$\mathbf{X}_{\boldsymbol{\tau}} = \left( \sum_{k=0}^{\infty} \frac{(-\tau_\eta)^k}{k!} \mathbf{L}_\eta^k \right) \mathbf{X}_1^M \left( \sum_{m=0}^{\infty} \frac{(-\tau_\xi)^m}{m!} \mathbf{L}_\xi^m \right). \tag{22}$$

Intuitively, for any nonzero $(\tau_\eta, \tau_\xi)$, this series propagates information from the whole graph, as no factor in front of the power of the Laplacian is zero. The truncated version of $\mathbf{X}_{\boldsymbol{\tau}}$ is defined by

$$\mathbf{X}_{\boldsymbol{\tau}}^{K_\eta, K_\xi} = \left( \sum_{k=0}^{K_\eta} \frac{(-\tau_\eta)^k}{k!} \mathbf{L}_\eta^k \right) \mathbf{X}_1^M \left( \sum_{m=0}^{K_\xi} \frac{(-\tau_\xi)^m}{m!} \mathbf{L}_\xi^m \right). \tag{23}$$

Therefore, we have

$$\left\| \mathbf{X}_{\boldsymbol{\tau}} - \mathbf{X}_{\boldsymbol{\tau}}^{K_\eta, K_\xi} \right\|$$

$$= \left\| \left( \sum_{k=0}^{\infty} \frac{(-\tau_\eta)^k}{k!} \mathbf{L}_\eta^k \right) \mathbf{X}_1^M \left( \sum_{m=0}^{\infty} \frac{(-\tau_\xi)^m}{m!} \mathbf{L}_\xi^m \right) - \left( \sum_{k=0}^{K_\eta} \frac{(-\tau_\eta)^k}{k!} \mathbf{L}_\eta^k \right) \mathbf{X}_1^M \left( \sum_{m=0}^{K_\xi} \frac{(-\tau_\xi)^m}{m!} \mathbf{L}_\xi^m \right) \right\|$$

$$= \left\| \left( \left( \sum_{k=0}^{K_\eta} \frac{(-\tau_\eta)^k}{k!} \mathbf{L}_\eta^k \right) + \left( \sum_{k=K_\eta+1}^{\infty} \frac{(-\tau_\eta)^k}{k!} \mathbf{L}_\eta^k \right) \right) \mathbf{X}_1^M \left( \sum_{m=0}^{\infty} \frac{(-\tau_\xi)^m}{m!} \mathbf{L}_\xi^m \right) - \right.$$

$$\left. \left( \sum_{k=0}^{K_\eta} \frac{(-\tau_\eta)^k}{k!} \mathbf{L}_\eta^k \right) \mathbf{X}_1^M \left( \sum_{m=0}^{K_\xi} \frac{(-\tau_\xi)^m}{m!} \mathbf{L}_\xi^m \right) \right\|$$

$$= \left\| \left( \sum_{k=K_\eta+1}^{\infty} \frac{(-\tau_\eta)^k}{k!} \mathbf{L}_\eta^k \right) \mathbf{X}_1^M \left( \sum_{m=0}^{\infty} \frac{(-\tau_\xi)^m}{m!} \mathbf{L}_\xi^m \right) + \right.$$

$$\left. \left( \sum_{k=0}^{K_\eta} \frac{(-\tau_\eta)^k}{k!} \mathbf{L}_\eta^k \right) \mathbf{X}_1^M \left( \sum_{m=K_\xi+1}^{\infty} \frac{(-\tau_\xi)^m}{m!} \mathbf{L}_\xi^m \right) \right\|$$

$$\leq \left\| \left( \sum_{k=K_\eta+1}^{\infty} \frac{(-\tau_\eta)^k}{k!} \mathbf{L}_\eta^k \right) \mathbf{X}_1^M \left( \sum_{m=0}^{\infty} \frac{(-\tau_\xi)^m}{m!} \mathbf{L}_\xi^m \right) \right\| +$$

$$\left\| \left( \sum_{k=0}^{K_\eta} \frac{(-\tau_\eta)^k}{k!} \mathbf{L}_\eta^k \right) \mathbf{X}_1^M \left( \sum_{m=K_\xi+1}^{\infty} \frac{(-\tau_\xi)^m}{m!} \mathbf{L}_\xi^m \right) \right\|$$

$$\leq \left( \left( \sum_{k=K_\eta+1}^{\infty} \frac{|\tau_\eta|^k}{k!} C_s^k \right) \left( \sum_{m=0}^{\infty} \frac{|\tau_\xi|^m}{m!} C_t^m \right) + \left( \sum_{k=0}^{\infty} \frac{|\tau_\eta|^k}{k!} C_s^k \right) \left( \sum_{m=K_\xi+1}^{\infty} \frac{|\tau_\xi|^m}{m!} C_t^m \right) \right) \|\mathbf{X}_1^M\|, \tag{24}$$

which completes the proof.

### B.2 PROOF OF THEOREM 1

Let $\mathbf{X}_0 \sim p_0$ and $\mathbf{X}_1 \sim q_1$ be the source and target distributions of a flow matching model, while $\mathbf{X}_1^M = \mathbf{X}_1 \circ \mathbf{M}$ being the observable data, then the transport cost $\mathcal{C}_{\text{FM}}(p_0 \to q_1)$ is defined as

follows:

$$\mathcal{C}_{\mathrm{FM}}(p_0 \rightarrow q_1) = \mathbb{E}_{\mathbf{X}_1 \sim p_1(\mathbf{X}_0)} \left[ \left\| \mathbf{X}_{t=0}(\mathbf{X}_1^M) - \mathbf{X}_1 \right\|^2 \right], \tag{25}$$

where $\mathbf{X}_{t=0}(\mathbf{X}_1^M)$ represents the source sample generated from the observable data $\mathbf{X}_1^M$. Specifically, for FM with a Gaussian prior, we have $\mathbf{X}_{t=0}^{Gauss}(\mathbf{X}_1^M) = \mathbf{X}_1^M + \mathbf{\Sigma} \circ (\mathbf{I} - \mathbf{M})$ with $\mathbf{\Sigma} \sim \mathcal{N}\left(\mathbf{0}, \sigma^2 \mathbf{I}\right)$ sampled from an isotropic Gaussian distribution. Therefore, we have

$$
\begin{aligned}
\mathcal{C}_{\mathrm{FM}}(p_0^{Gauss} \rightarrow q_1) &= \mathbb{E}_{\mathbf{X}_1 \sim p_1(\mathbf{X}_0)} \left[ \left\| \mathbf{X}_1^M + \mathbf{\Sigma} \circ (\mathbf{I} - \mathbf{M}) - \mathbf{X}_1 \right\|^2 \right] \\
&= \mathbb{E}_{\mathbf{X}_1 \sim p_1(\mathbf{X}_0)} \left[ \left\| \mathbf{X}_1 \circ \mathbf{M} + \mathbf{\Sigma} \circ (\mathbf{I} - \mathbf{M}) - \mathbf{X}_1 \right\|^2 \right] \\
&= \mathbb{E}_{\mathbf{X}_1 \sim p_1(\mathbf{X}_0)} \left[ \left\| (\mathbf{X}_1 - \mathbf{\Sigma}) \circ (\mathbf{I} - \mathbf{M}) \right\|^2 \right] \\
&= \mathbb{E}_{\mathbf{X}_1 \sim p_1(\mathbf{X}_0)} \left[ \left\| \mathbf{X}_1 \circ (\mathbf{I} - \mathbf{M}) \right\|^2 \right] + \mathbb{E}_{\mathbf{X}_1 \sim p_1(\mathbf{X}_0)} \left[ \left\| \mathbf{\Sigma} \circ (\mathbf{I} - \mathbf{M}) \right\|^2 \right] \\
&= \mathbb{E}_{\mathbf{X}_1 \sim p_1(\mathbf{X}_0)} \left[ \left\| \mathbf{X}_1 \circ (\mathbf{I} - \mathbf{M}) \right\|^2 \right] + \sigma^2 \mathrm{tr}\left(\mathbf{I} - \mathbf{M}\right).
\end{aligned}
\tag{26}
$$

For GiFlow, the graph-informed prior is constructed based on the spatiotemporal filtering defined in (5), leading to $\mathbf{X}_{t=0}^G(\mathbf{X}_1^M) = e^{-\tau_\eta \mathbf{L}_\eta} \mathbf{X}_1^M e^{-\tau_\xi \mathbf{L}_\xi}$. Since $\tau_\eta$ and $\tau_\xi$ are obtained by optimizing (6) with $\alpha_\tau = 0$, we have

$$\left\| \mathbf{X}_1 - e^{-\tau_\eta \mathbf{L}_\eta} \mathbf{X}_1^M e^{-\tau_\xi \mathbf{L}_\xi} \right\|^2 \leq \left\| \mathbf{X}_1 - \mathbf{X}_1^M \right\|^2 \tag{27}$$

since $\tau_\eta$ and $\tau_\xi$ represent the optimal solution of (6). Note that since Problem (6) is nonconvex, we can only obtain stationary solutions in practice. In such cases, we can optimize (6) using gradient-based methods with initialization $\tau_\eta = 0$ and $\tau_\xi = 0$, then Eq. (27) still holds. Therefore, we have

$$
\begin{aligned}
\mathcal{C}_{\mathrm{FM}}(p_0^G \rightarrow q_1) &= \mathbb{E}_{\mathbf{X}_1 \sim p_1(\mathbf{X}_0)} \left[ \left\| \mathbf{X}_1 - e^{-\tau_\eta \mathbf{L}_\eta} \mathbf{X}_1^M e^{-\tau_\xi \mathbf{L}_\xi} \right\|^2 \right] \\
&\leq \mathbb{E}_{\mathbf{X}_1 \sim p_1(\mathbf{X}_0)} \left[ \left\| \mathbf{X}_1 - \mathbf{X}_1^M \right\|^2 \right] \\
&= \mathbb{E}_{\mathbf{X}_1 \sim p_1(\mathbf{X}_0)} \left[ \left\| \mathbf{X}_1 - \mathbf{X}_1 \circ \mathbf{M} \right\|^2 \right] \\
&\leq \mathbb{E}_{\mathbf{X}_1 \sim p_1(\mathbf{X}_0)} \left[ \left\| \mathbf{X}_1 \circ (\mathbf{I} - \mathbf{M}) \right\|^2 \right] + \sigma^2 \mathrm{tr}\left(\mathbf{I} - \mathbf{M}\right).
\end{aligned}
\tag{28}
$$

Combining Eq. (26) and Eq. (28) completes the proof.

## C  EXPERIMENTS

### C.1  INTRODUCTION OF DATASETS

We conduct experiments on four real-world datasets: two air quality datasets, Air-36 and AQI, and two traffic datasets, PeMS04 and PeMS08. Air-36 and AQI collect hourly sampled PM2.5 pollutant data. Specifically, Air-36 is collected from 36 monitoring stations in Beijing, while AQI is collected from 437 monitoring stations spread across 43 Chinese cities. Both air quality datasets have 8760 timesteps, covering one year from 2014/05/01 to 2015/04/30 (Zheng et al., 2015). PeMS04 and PeMS08 are two traffic datasets about highway traffic flow in California, which are collected by the Caltrans Performance Measurement System (PeMS) (Chen et al., 2001). Specifically, PeMS04 is collected from 307 monitoring sensors covering two months from 2018/01/01 to 2018/02/28, while PeMS08 is collected from 170 monitoring sensors covering two months from 2016/07/01 to 2016/08/31. Both datasets originally collect data every 30 seconds, and the collected data is then aggregated with a 5-minute interval. The statistics of the datasets are summarized in Table 7.

### C.2  INTRODUCTION OF BASELINES

To evaluate the performance of our proposed method, we compare it with various baselines:

- **Mean-S**: imputes the missing values using the spatial average, i.e., the mean values of all nodes at a given timestep.

Table 7: Statistics of the datasets.

| Dataset | # Nodes | # Timesteps | Sampling intervals | Collected date |
|---------|---------|-------------|--------------------|----------------|
| Air-36  | 36      | 8760        | 1 hour             | 2014/05/01 – 2015/04/30 |
| AQI     | 437     | 8760        | 1 hour             | 2014/05/01 – 2015/04/30 |
| PeMS04  | 307     | 16992       | 5 minute           | 2018/01/01 – 2018/02/28 |
| PeMS08  | 107     | 17856       | 5 minute           | 2016/07/01 – 2016/08/31 |

- **Mean-T**: imputes the missing values using the temporal average, i.e., the mean values of all timesteps for a given node.
- **Linear**: imputes the missing values using temporal linear interpolation for each node independently.
- **KNN**: imputes the missing values using the average signal of neighboring nodes.
- **FP** (Rossi et al., 2022): performs feature propagation to impute the missing values.
- **BRITS** (Cao et al., 2018): a bidirectional RNN-based model.
- **SAITS** (Du et al., 2023): a transformer-based model.
- **SPIN** (Marisca et al., 2022): an efficient version of spatio-temporal attention-based method.
- **GRIN** (Cini et al., 2022): a GNN model with bidirectional gated recurrent unit.
- **OPCR** (Deng et al., 2024): a GNN model that contains attention-based one-step propagation and confidence-based refinement.
- **PriSTI** (Liu et al., 2023a): a conditional diffusion model.
- **ImputeFormer** (Nie et al., 2024): a low rankness-induced transformer model.
- **CoFILL** (He et al., 2025): a conditional diffusion model with a dual-stream architecture that processes temporal and frequency domain features in parallel.

## C.3 IMPLEMENTATION DETAILS

For all the experimental results, we give the average performance and standard deviation with 5 independent trials. For all the datasets, we select windows of length 24. For each dataset, we randomly select 70%/10%/20% of the data for training, validation, and testing. The Adam optimizer is used in all experiments for model training (Kingma & Ba, 2015). We fix the maximum number of epochs to 300, and we use early stopping on the validation set with a patience of 10 epochs. To stabilize the training process, we employed an exponential moving average (EMA) of the model parameters with a decay rate of 0.9999. To solve the ODE, we utilized an Euler solver with 20 steps. The models' hyperparameters are tuned based on the results of the validation set. The search space of hyperparameters are as follows: 1) learning rate: {0.005, 0.001, 0.0005}; 2) weight decay: {0, 5e-4, 5e-5, 5e-3}; 3) dropout rate: {0, 0.1, 0.2, 0.3}; 4) GNN layers: {2, 4, 6, 8}; 5) embedding dimension: {32, 64, 128}; 6) weight parameter $\alpha_\tau$: {0.1, 0.01, 0.001, 0.0001}.

## C.4 PERFORMANCE WITH DIFFERENT MISSING RATE

To evaluate the robustness of the model under different missing rates, we conduct experiments using the point missing strategy on the Air-36 dataset, with $\rho$ ranging from 20% to 60%. The results on MAE and MAPE are showcased in Figure 2. In the following, we present the results on RMSE in Figure 4. The results on RMSE exhibit similar patterns as in the MAE and MAPE results, where the imputation performance steadily degrades with increasing rates, and GiFlow consistently outperforms the other baselines across different missing rates.

## C.5 FILTERING FACTOR VALUES WITH INCREASING MISSING RATE

To validate the theoretical results presented in Proposition 1, we evaluate how the filtering factors $\tau_\eta$ and $\tau_\xi$ change with increasing missing rates. The results on the Air-36 dataset are showcased in Figure 3. In the following, we present the results on the AQI dataset in Figure 5. The patterns of

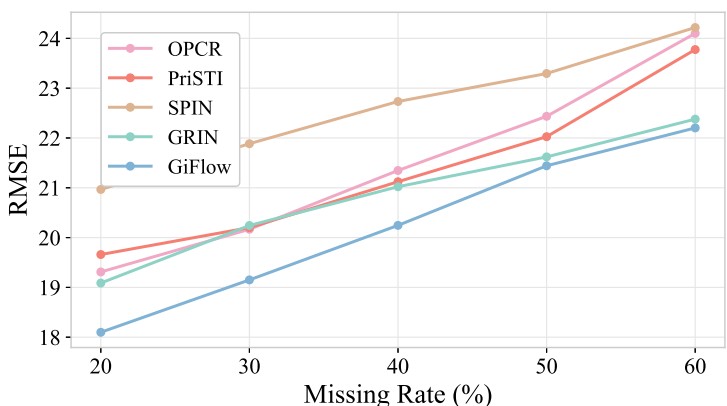

Figure 4: Performance on RMSE with different Missing Rate.

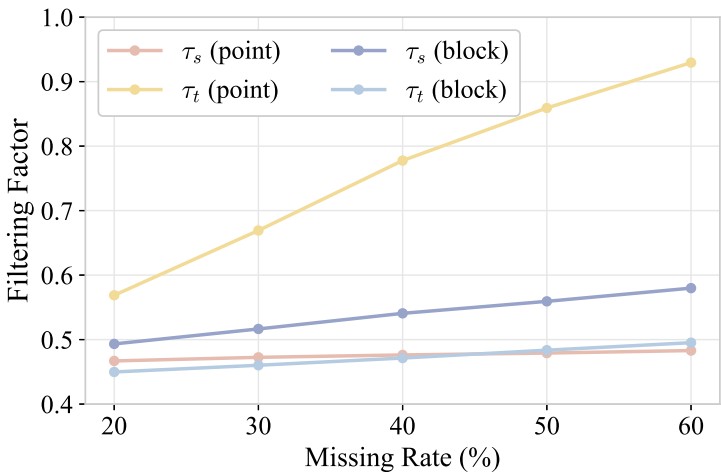

Figure 5: Filtering factor values with different missing rates on the AQI dataset.

$\tau_\eta$ and $\tau_\xi$ with increasing missing rate on the AQI dataset coincide with the patterns on the Air-36 dataset. Specifically, the filtering factors $\tau_\eta$ and $\tau_\xi$ increase as the missing rate increases, which corroborates the results we obtained from Proposition 1. Moreover, we observe that the increase of $\tau_\eta$ is more significant than the increase of $\tau_\xi$, indicating that the model relies more on spatial filtering than temporal filtering.

### C.6 IMPUTATION PERFORMANCE WITH BLOCK MISSING STRATEGY ON TRAFFIC DATASET

In this section, we conduct experiments using the block missing strategy with $\rho = 20\%$ on the PeMS08 dataset. The results are reported in Table 8. Similar as for the point missing setting, KNN and FP perform quite bad, validating that relying only on the spatial dependencies cannot characterize the system dynamics of traffic datasets well. The spatiotemporal methods still achieve good results, highlighting the importance of considering both spatial and temporal dependencies. The proposed GiFlow achieves the best results on all metrics, validating the applicability of GiFlow to other datasets under different missing patterns.

Table 8: Imputation performance with block missing strategy on traffic data ($\rho = 20\%$).

| Model | PeMS08 | | |
| --- | --- | --- | --- |
| | MAE | RMSE | MAPE |
| Mean-S | $34.69^{***}_{\pm 0.25}$ | $55.86^{***}_{\pm 0.67}$ | $32.39^{***}_{\pm 2.89}$ |
| Mean-T | $86.36^{***}_{\pm 0.47}$ | $113.37^{***}_{\pm 0.57}$ | $141.93^{***}_{\pm 3.10}$ |
| Linear | $32.17^{***}_{\pm 0.32}$ | $56.26^{***}_{\pm 0.47}$ | $34.51^{***}_{\pm 3.03}$ |
| KNN | $117.40^{***}_{\pm 1.13}$ | $152.28^{***}_{\pm 1.03}$ | $195.49^{***}_{\pm 6.17}$ |
| FP | $118.97^{***}_{\pm 0.44}$ | $149.08^{***}_{\pm 0.34}$ | $206.26^{***}_{\pm 5.25}$ |
| BRITS | $24.29^{***}_{\pm 0.25}$ | $38.36^{***}_{\pm 0.79}$ | $20.45^{***}_{\pm 1.62}$ |
| SAITS | $32.31^{***}_{\pm 1.82}$ | $47.23^{***}_{\pm 2.22}$ | $24.55^{***}_{\pm 2.27}$ |
| SPIN | $19.63^{*}_{\pm 0.16}$ | $30.43_{\pm 0.33}$ | $14.94_{\pm 1.85}$ |
| GRIN | $21.65^{***}_{\pm 0.41}$ | $\overline{36.28^{***}_{\pm 0.49}}$ | $16.42^{*}_{\pm 1.37}$ |
| OPCR | $22.96^{***}_{\pm 1.06}$ | $39.89^{***}_{\pm 1.13}$ | $21.70^{*}_{\pm 2.98}$ |
| PriSTI | $\underline{18.94}_{\pm 1.01}$ | $30.77_{\pm 1.32}$ | $\underline{14.48}_{\pm 0.54}$ |
| GiFlow | $\mathbf{18.70}_{\pm \mathbf{0.37}}$ | $\mathbf{29.97}_{\pm \mathbf{0.37}}$ | $\mathbf{14.02}_{\pm \mathbf{1.67}}$ |

## D  THE USE OF LARGE LANGUAGE MODELS

During the preparation of this work, the authors used ChatGPT to assist with grammar checking and text polishing. After using this tool, the authors carefully reviewed and edited the content as needed and take full responsibility for the content of this publication.

