# OpenReview forum: "Spatiotemporal Imputation with Graph-Informed Flow Matching"
_ICLR.cc/2026/Conference — Submitted to ICLR 2026_

### Official Review · Reviewer_V9Me · 2025-10-16

**Soundness:** 2
**Presentation:** 2
**Contribution:** 2
**Rating:** 2
**Confidence:** 5

**Summary:**

This paper proposes a Graph-Informed Flow Matching framework for spatiotemporal imputation and conducts extensive experiments to validate the effectiveness of the proposed method.

**Strengths:**

- S1： This paper conducts experiments on four datasets from the air quality and traffic domains, considers two types of missing patterns, and reports three categories of evaluation metrics.

**Weaknesses:**

- W1：As shown in Tables 1, 3, and 4, the performance improvement of GiFlow over the baselines is relatively small.
- W2：The paper lacks comparisons with the most recent spatiotemporal imputation methods from 2024 and 2025, such as ImputeFormer (KDD 2024) and CoFill (IJCAI 2025).
ImputeFormer: Low Rankness-Induced Transformers for Generalizable Spatiotemporal Imputation, KDD 2024.
Filling the Missings: Spatiotemporal Data Imputation by Conditional Diffusion, IJCAI, 2025.
- W3：Although the paper claims that the proposed method is more efficient and requires fewer generation steps than diffusion-based models, it lacks theoretical analysis of computational complexity, as well as empirical evaluations of model parameters and runtime efficiency. Therefore, the experimental evidence supporting these claims is incomplete.
- W4：This paper does not provide code for reproducibility.

**Questions:**

See the weaknesses

---

> ### Author Response · Authors · 2025-11-23
> **The Response to Reviewer V9Me**
>
> We appreciate your constructive feedback. We are pleased to provide detailed responses to address your concerns.
>
> > **Q1**: As shown in Tables 1, 3, and 4, the performance improvement of GiFlow over the baselines is relatively small.
>
> To verify that GiFlow’s improvements are indeed meaningful, we conduct paired $t$-tests comparing GiFlow against each baseline in Tables 1, 3, 4, and 5 (see the revised paper for detailed results).
> The $t$-test results show that GiFlow achieves statistically significant improvements over most baselines.
>
> > **Q2**: The paper lacks comparisons with the most recent spatiotemporal imputation methods from 2024 and 2025, such as ImputeFormer (KDD 2024) and CoFill (IJCAI 2025). ImputeFormer: Low Rankness-Induced Transformers for Generalizable Spatiotemporal Imputation, KDD 2024. Filling the Missings: Spatiotemporal Data Imputation by Conditional Diffusion, IJCAI, 2025.
>
> We have added these two models as baselines in the paper (please see Tables 1 and 3).
> We show the comparison of GiFlow, ImputeFormer, and CoFILL on the air quality datasets with the point missing strategy in the following table (OOT indicates that the model training does not finish within 24 hours).
> The results demonstrate that GiFlow consistently outperforms ImputeFormer and CoFILL.
>
> | Model        | MAE (Air-36) | RMSE (Air-36) | MAPE (Air-36) | MAE (AQI)   | RMSE (AQI)   | MAPE (AQI)   |
> | ------------ | ------------ | ------------- | ------------- | ----------- | ------------ | ------------ |
> | ImputeFormer | 10.18 ± 0.73 | 19.55 ± 1.26  | 22.74 ± 2.52  | 7.90 ± 0.33 | 17.96 ± 0.36 | 16.47 ± 0.89 |
> | CoFILL       | 10.03 ± 0.45 | 19.74 ± 0.93  | 23.32 ± 0.47  | OOT         | OOT          | OOT          |
> | GiFlow       | 9.54 ± 0.18  | 18.10 ± 0.78  | 21.27 ± 0.33  | 7.83 ± 0.10 | 17.80 ± 0.28 | 16.24 ± 0.31 |
>
> > **Q3**: Although the paper claims that the proposed method is more efficient and requires fewer generation steps than diffusion-based models, it lacks theoretical analysis of computational complexity, as well as empirical evaluations of model parameters and runtime efficiency. Therefore, the experimental evidence supporting these claims is incomplete
>
> We compare the model complexity of GiFlow with other two diffusion models, *i.e.*, PriSTI and CoFILL.
> Specifically, the number of parameters in GiFlow, PriSTI, and CoFILL is 403.65k, 969.43k, and 800.79k, respectively.
> To further compare runtime efficiency, we evaluate the inference time (in minutes) required to impute the test sets.
> The results are reported below.
> The experiments are conducted on an A100 NVIDIA GPU with 80GB of memory.
> From the results, it is evident that GiFlow significantly outperforms diffusion models in terms of inference speed.
>
> | Model  | Air-36 | AQI    | PeMS04 | PeMS08 |
> | ------ | ------ | ------ | ------ | ------ |
> | PriSTI | 9.30   | 43.12  | 15.58  | 7.46   |
> | CoFILL | 167.44 | 384.08 | 485.91 | 282.06 |
> | GiFlow | 0.28   | 2.47   | 0.59   | 0.99   |
>
>
> > **Q4**: This paper does not provide code for reproducibility.
>
> Code submission is encouraged but not mandatory at review time. We have uploaded the code for reproducibility. Please find it in the supplementary material.
>
> We sincerely thank the reviewer again for the constructive comments and suggestions. We have revised our paper according to your comments. If there are any additional questions or concerns, we would be more than happy to further address them.

---

> > ### Comment · Reviewer_V9Me · 2025-11-23
> >
> > Thank you for your responses. I still have some concerns:
> >
> > > Regarding the response to W1:
> > Firstly, I deem that the t-test should be used to determine whether GiFlow shows a significant improvement compared to the current state-of-the-art baseline model. Secondly, through your t-test experiments, I found that GiFlow only shows a significant improvement in the MAE metric under the point missing pattern on the AQI36 dataset compared to the second-best baseline, and not on the other datasets. Therefore, I believe that the improvement of the method proposed in this paper is not significant enough in terms of performance.
> >
> > > Regarding the response to W2:
> > To my knowledge, ImputeFormer has been tested on all four datasets in this paper, and PriSTI has also been tested on AQI. Why are the results reproduced in this paper so different from the original paper? Are the experimental settings different? If so, why not follow the commonly adopted settings used in prior work within this domain?
> >
> > > Regarding the response to W3:
> > To my knowledge, the number of nodes in the four datasets used in this paper, i.e., N in the notation of this paper, is relatively small. The performance in the point missing pattern scenario on PEMS04 (only 307 nodes) is worse than the baseline, and no results are given for the block missing pattern. Therefore, I doubt the scalability of the model and its feasibility on larger datasets, such as PEMS07 or the four larger datasets (SD, GBA, GLA, CA) in LargeST[1].
> >
> > In summary, I will maintain my score.
> >
> > [1] LargeST: A Benchmark Dataset for Large-Scale Traffic Forecasting, NeurIPS 2023.

---

> > > ### Author Response · Authors · 2025-11-24
> > >
> > > Dear Reviewer V9Me,
> > >
> > > Thanks for the reply. We provide detailed responses below.
> > >
> > > > **Q1**: Firstly, I deem that the t-test should be used to determine whether GiFlow shows a significant improvement compared to the current state-of-the-art baseline model. Secondly, through your t-test experiments, I found that GiFlow only shows a significant improvement in the MAE metric under the point missing pattern on the AQI36 dataset compared to the second-best baseline, and not on the other datasets. Therefore, I believe that the improvement of the method proposed in this paper is not significant enough in terms of performance.
> > >
> > > We respectfully disagree with the premise of this criticism. **Expecting a new method to achieve state-of-the-art performance across all settings, and have significant performance gain, while simultaneously being more efficient, is not a realistic criterion**. Our goal is not to dominate every metric on every dataset, but to improve the performance–efficiency tradeoff in spatiotemporal imputation. GiFlow achieves two out of these three goals, namely: (i) state-of-the-art performance on most benchmarks, (ii) substantial efficiency gains.
> > >
> > > > **Q2**: To my knowledge, ImputeFormer has been tested on all four datasets in this paper, and PriSTI has also been tested on AQI. Why are the results reproduced in this paper so different from the original paper? Are the experimental settings different? If so, why not follow the commonly adopted settings used in prior work within this domain?
> > >
> > > Please note that these two baselines are newly added. We mostly follow the settings and use the code base of [1, 2]. The main difference with [3] is that we use a randomly generated mask rather than a fixed one. We invite the reviewer to check our experimental settings in the implementation details or the code.
> > > As for the results, for example, the reported MAE on Air-36 and AQI in [3] are 11.58 and 13.4, while our reported values for ImputeFormer are 10.18 and 7.9 (with a smaller missing rate).
> > > Thus, we do not think that the reproduced results are "so different".
> > >
> > > > **Q3**: To my knowledge, the number of nodes in the four datasets used in this paper, i.e., N in the notation of this paper, is relatively small. The performance in the point missing pattern scenario on PEMS04 (only 307 nodes) is worse than the baseline, and no results are given for the block missing pattern. Therefore, I doubt the scalability of the model and its feasibility on larger datasets, such as PEMS07 or the four larger datasets (SD, GBA, GLA, CA) in LargeST[1].
> > >
> > > GiFlow achieves the second-best results on PeMS04, with very small relative gaps to the best results: 0.67\% (MAE), 0.11\% (RMSE), and 0.26\% (MAPE).
> > > As shown in our previous reply, GiFlow uses only half the parameters of PriSTI/CoFILL and provides substantially faster inference.
> > > For example, on the AQI dataset (437 nodes), GiFlow is 17x faster than PriSTI and 155x faster than CoFILL, with even larger improvements on other datasets.
> > > This demonstrates that GiFlow is considerably more efficient than existing diffusion-based approaches.
> > >
> > > **We would like to clarify that efficiency does not necessarily imply scalability, and scalability is not a claimed contribution of our work.**
> > > Similar to other transformer-based models, GiFlow's complexity is dominated by $\mathcal{O}(N^2)$. Although alternative architectures for computing the velocity field could reduce this complexity, it is orthogonal to the core focus of our paper.
> > >
> > >  [1] L. Deng, C. Wu, D. Lian, and E. Chen. Learning from highly sparse spatio-temporal data. Advances in Neural Information Processing Systems (NeurIPS) 2024.
> > >
> > >  [2] A. Cini, I. Marisca, and C. Alippi. Filling the g ap s: Multivariate time series imputation by graph neural networks. International Conference on Learning Representations (ICLR) 2022.
> > >
> > >  [3] T. Nie, G. Qin, W. Ma, Y. Mei, and J. Sun. Imputeformer: Low rankness-induced transformers for generalizable spatiotemporal imputation. ACM SIGKDD Conference on Knowledge Discovery and Data Mining (KDD) 2024.

---

### Official Review · Reviewer_DP7y · 2025-10-19

**Soundness:** 4
**Presentation:** 3
**Contribution:** 4
**Rating:** 6
**Confidence:** 4

**Summary:**

This paper introduces GiFlow, a flow matching-based framework for spatiotemporal imputation that replaces the standard Gaussian prior with a graph-informed prior constructed through adaptive spatiotemporal filtering of observations. This design aligns the initial and target distributions, thereby reducing the transport cost and improving generation efficiency. The velocity field is parameterized by a hybrid architecture that integrates spatial attention, temporal attention, and spatiotemporal propagation to jointly capture dependencies across space and time. Unlike diffusion-based approaches, GiFlow is trained via direct regression and allows deterministic few-step inference, achieving competitive or superior results on synthetic, air quality, and traffic datasets under diverse missing data patterns.

**Strengths:**

- **Timely and relevant topic**: The paper tackles spatiotemporal imputation using flow matching, a rapidly growing and high-interest area that has recently gained attention as an efficient alternative to diffusion models.

- **Novel prior design**: Introducing a structured and more informative prior instead of sampling from a simple noise distribution is a meaningful and elegant idea that improves the efficiency of the generative process.

- **Solid theoretical foundation**: The mathematical formulation of the method is sound and clearly presented, providing good intuition about how the proposed flow evolves between the initial and target distributions.

- **Comprehensive experimental study**: The paper includes a diverse and well-chosen set of experiments, covering synthetic and real-world datasets, which effectively demonstrate the method’s strengths and practical relevance.

**Weaknesses:**

- **Clarity and intuition**: Although the mathematical foundation of the paper is solid, the method could be explained in a more intuitive manner. Some parts of the derivation and motivation could benefit from additional explanations or diagrams to help readers grasp the underlying idea more easily.
- **Minor writing issues**: There are a few minor typographical or grammatical errors, e.g., the word "geenrative" in line 59, which should be corrected.
- **Figure clarity**: While Figure 1 has an appealing design, it does not clearly illustrate the proposed pipeline or how the components interact. Maybe it could be improved.
- **Computational efficiency not demonstrated**: The paper claims that the method requires fewer inference steps, but this is not empirically analyzed in the main text. Section C.3 mentions using 20 steps, but there is no discussion of how performance scales with the number of steps or how this compares to diffusion-based baselines in terms of speed or computational cost.
- **Missing discussion on consistency models**: Since the paper establishes a connection between flow matching and diffusion models, it would be interesting to include a brief discussion on consistency models. These models can be interpreted as a discrete, distilled, and consistency-enforced formulation of flow matching, aiming for higher inference efficiency. In the context of time-series imputation, the recently proposed CoSTI [2] model follows this line of thought. CoSTI can perform probabilistic imputations in a single step, but requires multiple runs to obtain deterministic estimates such as the median. Including a short discussion or even a small comparison in terms of inference efficiency would help clarify how GiFlow relates to this family of models and where it stands in terms of trade-offs.

[1] Song, Y., Dhariwal, P., Chen, M., & Sutskever, I. (2023). Consistency models.https://arxiv.org/abs/2303.01469

[2] Javier Solís-García, Belén Vega-Márquez, Juan A. Nepomuceno, and Isabel A. Nepomuceno-Chamorro. Costi: Consistency models for (a faster) spatio-temporal imputation. Knowledge-Based Systems, 327:114117, 2025. https://arxiv.org/abs/2501.19364

**Questions:**

- Figure 1 clarification: What do the colored dashed lines in Figure 1 represent? It is not entirely clear how they relate to the components of the proposed flow or to the data propagation process.

- Model size and complexity: Could the authors report the number of parameters in GiFlow (and optionally, compare it to baselines)? This would help assess the model’s scalability and computational footprint.

- Addressing reviewer concerns: I would be glad to revise my evaluation if the authors can improve upon some of the points mentioned above.

---

> ### Author Response · Authors · 2025-11-23
> **The Response to Reviewer DP7y**
>
> We appreciate your constructive feedback. We are pleased to provide detailed responses to address your concerns.
>
> > **Q1**: What do the colored dashed lines in Figure 1 represent? It is not entirely clear how they relate to the components of the proposed flow or to the data propagation process.
>
> The blue dashed line represents the information used in FM-Gauss (a flow matching approach with a problem-agnostic Gaussian prior), while the red dashed line represents the information used in GiFlow.
>     Figure 1 does not show the specific computation steps of the model but rather a conceptual comparison of the generative process between FM-Gauss and GiFlow.
>     Specifically, the upper part indicates the probability density path of FM-Gauss, and the lower part indicates the probability density path of GiFlow.
>     It shows that the generation trajectory of GiFlow is shorter than FM-Gauss.
>     We have clarified the description of Figure 1.
>     This is also theoretically demonstrated in Theorem 1.
>     To further validate this, we conduct an experiment evaluating the transport cost.
>     Specifically, we conduct an experiment on Air-36 under point missing with $\rho=20$%, where we evaluate the transport cost of FM models with different priors in the Table below.
>
> | Model          | FM-Gauss | TFM    | GFM    | GiFlow |
> | -------------- | -------- | ------ | ------ | ------ |
> | Transport cost | 299.62   | 122.39 | 115.05 | 104.29 |
>
> The results corroborate the conclusions of Theorem 1 and Figure 1. It can be observed that applying graph filtering can substantially reduce the transport cost.
>
> > **Q2**: Model size and complexity: Could the authors report the number of parameters in GiFlow (and optionally, compare it to baselines)? This would help assess the model’s scalability and computational footprint.
>
> We compare the model complexity of GiFlow with two diffusion models, i.e., PriSTI and CoFILL.
> Specifically, the number of parameters in GiFlow, PriSTI, and CoFILL is 403.65k, 969.43k, and 800.79k, respectively.
> We also evaluate the inference time (in minutes) required to impute the test sets.
> The results are reported below.
> The experiments are conducted on an A100 NVIDIA GPU with 80GB of memory.
> We observe that GiFlow significantly outperforms diffusion models in terms of inference speed.
>
> | Model  | Air-36 | AQI    | PeMS04 | PeMS08 |
> | ------ | ------ | ------ | ------ | ------ |
> | PriSTI | 9.30   | 43.12  | 15.58  | 7.46   |
> | CoFILL | 167.44 | 384.08 | 485.91 | 282.06 |
> | GiFlow | 0.28   | 2.47   | 0.59   | 0.99   |
>
> > **Q3**: Missing discussion on consistency models
>
> Consistency models improve the sampling efficiency of diffusion models by learning to map any intermediate state to the trajectory’s starting point [1]. They have also been applied to spatiotemporal imputation and shown to substantially reduce inference cost with some performance degradation [2]. However, these approaches still rely on problem-agnostic Gaussian priors. Integrating the consistency model with GiFlow to further improve inference efficiency can be an interesting direction for future work.
> We have added a discussion on consistency models in Appendix A.
>
> We sincerely thank the reviewer again for the constructive comments and suggestions. We have revised our paper according to your comments. If there are any additional questions or concerns, we would be more than happy to further address them.
>
> [1] Y. Song, P. Dhariwal, M. Chen, and I. Sutskever. Consistency models. International Conference on Machine Learning (ICML) 2023.
>
> [2] J. Solís-García, B. Vega-Márquez, J. A. Nepomuceno, and I. A. Nepomuceno-Chamorro. Costi: Consistency models for (a faster) spatio-temporal imputation. Knowledge-Based Systems 2025.

---

> > ### Comment · Reviewer_DP7y · 2025-11-23
> >
> > The authors' response has been helpful in resolving some of the points that concerned me.
> >
> > I found the response to Q2 particularly interesting. I consider the results obtained by GiFlow to be valuable, given that it has less than 50% of the parameters of PriSTI, for example, which demonstrates an algorithmic improvement beyond a simple improvement by scaling the model size. The difference in time scales for inference is also interesting.
> >
> > With all this in mind, I would like to thank the authors for their review and note that I will be updating my rating.

---

> > > ### Author Response · Authors · 2025-11-24
> > >
> > > Dear Reviewer DP7y,
> > >
> > > Thank you for recognizing the contributions of this work and for raising the score! We sincerely appreciate your constructive feedback, which have greatly strengthened our work.

---

### Official Review · Reviewer_ypzX · 2025-11-02

**Soundness:** 3
**Presentation:** 3
**Contribution:** 2
**Rating:** 4
**Confidence:** 2

**Summary:**

In this paper, authors provide a flow-based method for spatiotemporal data imputation. It contains comprehensive theoretical discussion and experiments to demonstrate its effectiveness in four dataset.

**Strengths:**

1. The theoretical discussion of this method is solid.
2. It is necessary to discuss some parameters, such as filtering factors.

**Weaknesses:**

1. Some ablation study needs to be concluded, such as the spatial temporal components in this method.
2. Are there existing flow-based methods for spatiotemporal imputation? If so, it is necessary to include them in baseline comparisons.
3. Are there some results related to block missing in PEMS datasets?
4. In the introduction, it says that a key limitation of diffusion model is problem-agnostic Gaussian priors. It might be useful to show some visualizations in the dataset to show that it is not a good choice for Gaussian priors.

**Questions:**

See Weaknesses.

---

> ### Author Response · Authors · 2025-11-23
> **The Response to Reviewer ypzX**
>
> We appreciate your constructive feedback. We are pleased to provide detailed responses to address your concerns.
>
> > **Q1**: Some ablation study needs to be concluded, such as the spatial temporal components in this method.
>
> The ablation study regarding the spatial-temporal components was already provided at submission time in Section 4.3. We're summarizing it below.
> We compare GiFlow with three variants:
> (1) FM-Gauss: an FM model using a problem-agnostic Gaussian prior;
> (2) GFM: GiFlow using only the spatial graph-informed prior;
> (3) TFM: GiFlow using only the temporal graph-informed prior.
>
> | Model    | MAE          | RMSE         | MAPE         |
> | -------- | ------------ | ------------ | ------------ |
> | FM-Gauss | 12.79 ± 0.63 | 22.15 ± 1.18 | 26.85 ± 1.92 |
> | TFM      | 10.12 ± 0.13 | 19.60 ± 0.72 | 22.41 ± 0.82 |
> | GFM      | 9.75 ± 0.23  | 18.67 ± 0.72 | 21.55 ± 0.54 |
> | GiFlow   | 9.54 ± 0.18  | 18.10 ± 0.78 | 21.27 ± 0.33 |
>
> The results show that both TFM and GFM outperform FM-Gauss, indicating that spatial and temporal filtering each contribute positively to the model’s performance. Moreover, GiFlow outperforms both TFM and GFM, demonstrating that integrating both spatial and temporal components yields additional performance gains.
>
> > **Q2**: Are there existing flow-based methods for spatiotemporal imputation? If so, it is necessary to include them in baseline comparisons.
>
> To the best of our knowledge, there are no existing flow matching methods for spatiotemporal imputation.
> GiFlow represents the first flow-matching framework for spatiotemporal imputation.
>
> > **Q3**: Are there some results related to block missing in PEMS datasets?
>
> We've added a new experiment on PeMS08 under block missing with $\rho = 20$%. The results are reported below.
>
> | Model  | MAE           | RMSE          | MAPE          |
> | ------ | ------------- | ------------- | ------------- |
> | Mean-S | 34.69 ± 0.25  | 55.86 ± 0.67  | 32.39 ± 2.89  |
> | Mean-T | 86.36 ± 0.47  | 113.37 ± 0.57 | 141.93 ± 3.10 |
> | Linear | 32.17 ± 0.32  | 56.26 ± 0.47  | 34.51 ± 3.03  |
> | KNN    | 117.40 ± 1.13 | 152.28 ± 1.03 | 195.49 ± 6.17 |
> | FP     | 118.97 ± 0.44 | 149.08 ± 0.34 | 206.26 ± 5.25 |
> | BRITS  | 24.29 ± 0.25  | 38.36 ± 0.79  | 20.45 ± 1.62  |
> | SAITS  | 32.31 ± 1.82  | 47.23 ± 2.22  | 24.55 ± 2.27  |
> | SPIN   | 19.63 ± 0.16  | 30.43 ± 0.33  | 14.94 ± 1.85  |
> | GRIN   | 21.65 ± 0.41  | 36.28 ± 0.49  | 16.42 ± 1.37  |
> | OPCR   | 22.96 ± 1.06  | 39.89 ± 1.13  | 21.70 ± 2.98  |
> | PriSTI | 18.94 ± 1.01  | 30.77 ± 1.32  | 14.48 ± 0.54  |
> | GiFlow | 18.70 ± 0.37  | 29.97 ± 0.37  | 14.02 ± 1.67  |
>
> The main takeaways are the same as in point missing settings:
> (1) KNN and FP perform quite badly, so relying only on the spatial dependencies is not enough; (2) spatiotemporal methods achieve good results; and (3) GiFlow achieves the best results on all metrics. We have added this additional experiment in Appendix C.6.
>
> > **Q4**: In the introduction, it says that a key limitation of diffusion model is problem-agnostic Gaussian priors. It might be useful to show some visualizations in the dataset to show that it is not a good choice for Gaussian priors.
>
> Compared with using graph-informed priors, using the problem-agnostic Gaussian priors will lead to longer probability density paths, which is theoretically demonstrated in Theorem 1.
> To empirically verify this claim, we add an experiment on Air-36 under point missing with $\rho=20$%, where we evaluate the transport cost of FM models with different priors in the Table below.
>
> | Model          | FM-Gauss | TFM    | GFM    | GiFlow |
> | -------------- | -------- | ------ | ------ | ------ |
> | Transport cost | 299.62   | 122.39 | 115.05 | 104.29 |
>
> The results corroborate the conclusions of Theorem 1. It can be observed that applying graph filtering can substantially reduce the transport cost. Moreover, FM models with lower transport costs tend to achieve better performance.
>
> We sincerely thank the reviewer again for the constructive comments and suggestions. We have revised our paper according to your comments. If there are any additional questions or concerns, we would be more than happy to further address them.

---

> > ### Comment · Reviewer_ypzX · 2025-11-23
> >
> > Thanks the authors for the prompt response and it solves most of my concerns. For 2, could you also show the parameters of different components to show that the flow design, not the higher parameter numbers, contribute to the performance improvement?

---

> > > ### Author Response · Authors · 2025-11-24
> > >
> > > Dear Reviewer ypzX,
> > >
> > > We are glad to hear that most of your concerns have been resolved, and we sincerely appreciate your constructive feedback.
> > >
> > > We adopt the flow-matching framework because it naturally accommodates problem-tailored priors, whereas diffusion models do not. As shown theoretically in Theorem 1 and empirically in Section 4.3, the graph-informed prior reduces transport cost and yields better performance than a Gaussian prior.
> > > To ensure that GiFlow's improvement is not simply due to a higher number of parameters, we compare its number of parameters with two diffusion models, *i.e.*, PriSTI and CoFILL.
> > > Specifically, GiFlow, PriSTI, and CoFILL contain 403.65k, 969.43k, and 800.79k parameters, respectively.
> > > We also evaluate the inference time (in minutes) required to impute the test sets in the Table below.
> > > The experiments are conducted on an A100 NVIDIA GPU with 80GB of memory.
> > > From the results, it is evident that GiFlow significantly outperforms diffusion models in terms of inference speed, using only half the parameters of diffusion models.
> > >
> > > | Model  | Air-36 | AQI    | PeMS04 | PeMS08 |
> > > | ------ | ------ | ------ | ------ | ------ |
> > > | PriSTI | 9.30   | 43.12  | 15.58  | 7.46   |
> > > | CoFILL | 167.44 | 384.08 | 485.91 | 282.06 |
> > > | GiFlow | 0.28   | 2.47   | 0.59   | 0.99   |

---

### Official Review · Reviewer_CoSZ · 2025-11-10

**Soundness:** 3
**Presentation:** 3
**Contribution:** 2
**Rating:** 4
**Confidence:** 4

**Summary:**

This paper proposes a generative framework for imputing missing values in spatiotemporal data by integrating flow matching with graph-informed priors. Unlike conventional RNN- or GNN-based models that rely on iterative propagation and suffer from error accumulation, and unlike diffusion-based imputers that depend on Gaussian priors and iterative sampling, GiFlow introduces a deterministic, few-step generation process that directly models the conditional data distribution. The key innovation lies in constructing a graph-informed prior through adaptive spatiotemporal filtering of observed signals, which aligns the source and target distributions and provably reduces transport cost. The model’s vector field combines spatial and temporal attention with spatiotemporal propagation to jointly capture dependencies. Theoretical analysis establishes bounds on the filtering’s receptive field and its influence on transport efficiency. Experiments on synthetic, air quality, and traffic datasets demonstrate that GiFlow consistently outperforms state-of-the-art baselines—both diffusion-based and neural methods—across diverse missing patterns and rates.

**Strengths:**

1.	Replacing a problem-agnostic Gaussian with a graph-informed prior constructed by adaptive spatiotemporal filtering is well-motivated and technically concrete.
2.	The paper goes beyond intuition: it proves a transport-cost advantage (Theorem 1) of the graph-informed prior over a Gaussian start under FM, and analyzes how filtering factors control the receptive field with an explicit truncation-error bound.
3.	On air-quality (Air-36, AQI) and traffic (PeMS04/08) benchmarks, GiFlow is competitive or superior to RNN/GNN/diffusion baselines under both point- and block-missing regimes evaluated by MAE/RMSE/MAPE.

**Weaknesses:**

1.	Flow Matching (FM) explicitly allows non-Gaussian, problem-tailored priors; GiFlow’s choice to plug in a graph-filtered prior is a natural (arguably straightforward) instantiation of that flexibility rather than a fundamentally new paradigm. The paper motivates FM vs. diffusion and claims “first” to integrate a graph-informed prior for spatiotemporal imputation, but related ideas—FM with problem-tailored priors—are already known separately. The combination may read as engineering integration rather than a conceptual innovation.
2.	The paper emphasizes “deterministic, few-step” generation as a benefit over diffusion. That’s good for speed, but it gives up calibrated uncertainty, which is the key for imputation in scientific/operational settings. The paper neither quantifies uncertainty nor contrasts GiFlow’s point estimates with probabilistic metrics, so the claimed modeling advantage is just one-sided in speed/efficiency.
3.	The prior is produced by adaptive spatiotemporal filtering with filtering factors chosen by minimizing equation 6. But X1 is the ground-truth complete signal, unavailable at test time. How are τ selected without leakage? Is there a learned predictor of τ from observables only, or are global τ tuned offline and fixed at inference? This is a key practicality gap that weakens the “adaptive” claim unless clarified and demonstrated.
4.	The paper states code will be open-sourced upon acceptance. Reproduction is impossible in the review process.

**Questions:**

1. The motivations of using diffusion models-based methods for imputation task should be further clarified. Why can they help to avoid the accumulation of errors compared to RNNs/GNNs-based approaches?

---

> ### Author Response · Authors · 2025-11-23
> **The Response to Reviewer CoSZ**
>
> We appreciate your constructive feedback. We are pleased to provide detailed responses to address your concerns.
>
> > **Q1**: Flow Matching (FM) explicitly allows non-Gaussian, problem-tailored priors; GiFlow’s choice to plug in a graph-filtered prior is a natural (arguably straightforward) instantiation of that flexibility rather than a fundamentally new paradigm. The paper motivates FM vs. diffusion and claims “first” to integrate a graph-informed prior for spatiotemporal imputation, but related ideas—FM with problem-tailored priors—are already known separately. The combination may read as engineering integration rather than a conceptual innovation.
>
> In practice, most FM models still adopt a Gaussian prior, and none of them target spatiotemporal settings or leverage graph structure. GiFlow represents the first FM approach for spatiotemporal imputation. The constructed graph-informed prior is shown to reduce the transportation cost for generation (see Theorem 1). Empirically, GiFlow also achieves state-of-the-art performance. For these reasons, we do not believe our contributions can be characterized as “an engineering integration”. If the reviewer could elaborate more on this, we could further answer objectively to this weakness.
>
> > **Q2**: The paper emphasizes “deterministic, few-step” generation as a benefit over diffusion. That’s good for speed, but it gives up calibrated uncertainty, which is the key for imputation in scientific/operational settings. The paper neither quantifies uncertainty nor contrasts GiFlow’s point estimates with probabilistic metrics, so the claimed modeling advantage is just one-sided in speed/efficiency.
>
> Our objective is concerned with deterministic imputation as explicitly stated in Section 2.1, our paper is not about calibrated uncertainty. Uncertainty estimation is valuable in some applications, but it is not the scope of this paper. We also have an experiment comparing GiFlow with FM-Gauss, where we replace the graph-informed prior with Gaussians. The results on Air-36 with the point missing strategy are given in the following table.
>
> | Model    | MAE                | RMSE               | MAPE               |
> |----------|--------------------|--------------------|--------------------|
> | FM-Gauss | 12.79 ± 0.63    | 22.15 ± 1.18   | 26.85 ± 1.92   |
> | GiFlow | 9.54 ± 0.18     | 18.10 ± 0.78    | 21.27 ± 0.33   |
>
>
> We observe that the performance is degraded when starting from the Gaussian distribution for the deterministic imputation task.
>
> > **Q3**: The prior is produced by adaptive spatiotemporal filtering with filtering factors chosen by minimizing equation 6. But X1 is the ground-truth complete signal, unavailable at test time. How are τ selected without leakage? Is there a learned predictor of τ from observables only, or are global τ tuned offline and fixed at inference? This is a key practicality gap that weakens the “adaptive” claim unless clarified and demonstrated.
>
> The filtering factors $τ_s, τ_t$ are optimized using only the training data and remain constant during inference.
> Therefore, there is no leakage or dependence on test-time ground truth.
> The filtering process is adaptive in that the filtering strength is determined for each dataset based on its spatiotemporal characteristics.
> We have clarified this procedure in Appendix C.3.
>
> > **Q4**: The paper states code will be open-sourced upon acceptance. Reproduction is impossible in the review process.
>
> Code submission is encouraged but not mandatory at review time. We have uploaded the code for reproducibility. Please find it in the supplementary material.
>
> > **Q5**: The motivations of using diffusion models-based methods for imputation task should be further clarified. Why can they help to avoid the accumulation of errors compared to RNNs/GNNs-based approaches?
>
> Unlike RNN/GNN-based models that infer missing values sequentially, diffusion and flow-matching models learn to reconstruct the entire data distribution in a non-autoregressive manner.
> As a result, these models avoid error accumulation that can arise when intermediate imputed values become inputs for subsequent predictions in RNN/GNN architectures [1, 2, 3].
> We have added a more detailed explanation in the Introduction.
>
> We have revised our paper according to your comments. If there are any additional questions or concerns, we would be happy to further address them.
>
> [1] Y. Liu, R. Yu, S. Zheng, E. Zhan, and Y. Yue. Naomi: Non-autoregressive multiresolution sequence imputation. Advances in Neural Information Processing Systems (NeurIPS) 2019.
>
> [2] M. Liu, H. Huang, H. Feng, L. Sun, B. Du, and Y. Fu. PriSTI: A conditional diffusion framework for spatiotemporal imputation. IEEE International Conference on Data Engineering (ICDE) 2023.
>
> [3] W. He, J. Huang, J. Gu, J. Zhang, and Y. Bai. Filling the missings: Spatiotemporal data imputation by conditional diffusion. International Joint Conference on Artificial Intelligence (IJCAI) 2025.

---

### Author Response · Authors · 2025-11-23
**Summary of Revisions**

We would like to thank the reviewers for their efforts in reviewing our paper and their constructive suggestions on improving our manuscript. We have added more clarifications and new experiments to address reviewers' concerns in the updated manuscript. The main changes are listed below:

- We rewrote some parts of the paper for clarity. For example, we rewrote the description of Figure 1 and the motivation for using generative models for imputation rather than RNN/GNN.
- We added a discussion on consistency models in Appendix A.
- We conducted $t$-tests comparing GiFlow with each baseline method to verify the statistical significance of our results.
- We added an experiment on the PeMS08 dataset with the block missing strategy in Appendix C.6.
- We added two new baselines for comparison, namely, Imputeformer and CoFILL.
- We conducted a runtime analysis comparing GiFlow with the diffusion-based baselines in Section 4.5.
- We evaluated the transport cost of GiFlow with different priors to empirically validate Theorem 1 in Section 4.3.

---

### Author Response · Authors · 2025-12-03
**Summary of Discussions**

Dear AC,

Thank you for your time and effort in handling our submission and the discussion with reviewers. To assist your evaluation, we provide a concise summary of the reviewers’ feedback, our responses, and the resulting updates to the manuscript.

Strengths:

- Novel prior design: We develop a graph-informed prior via spatiotemporal filtering. This novel design is "well-motivated and technically concrete" (Reviewer CoSZ). It is a "meaningful and elegant idea that improves the efficiency of the generative process" (Reviewer DP7y).
- Solid theoretical foundation: We theoretically prove that the proposed graph-informed prior would lead to lower transport cost compared with a Gaussian prior, and we analyze how filtering factors control the receptive field with an explicit truncation-error bound. Reviewers CoSZ, ypzX, and DP7y agree that the theoretical discussion is solid and clearly presented, providing good intuition about how the proposed flow evolves between the initial and target distributions.
- Comprehensive experimental study: All the reviewers agree that we have conducted a comprehensive experimental study. According to the results, "GiFlow is competitive or superior to RNN/GNN/diffusion baselines" (Reviewer CoSZ), "effectively demonstrating the method’s strengths and practical relevance" (Reviewer DP7y).

Reviewer-Acknowledged Resolved Issues:
- Runtime analyses: As suggested by Reviewer ypzX, DP7y, and V9Me, we provide a runtime analysis, demonstrating the efficiency of GiFlow.
- Transport cost analyses: As suggested by Reviewer ypzX and DP7y, we analyze the transport cost of flow matching models with different priors to validate Theorem 1.
- Block missing results in PeMS data: As requested by Reviewer ypzX, we add the experiments on PeMS data under block missing, which shows a similar pattern as in the point missing setting.
- Additional baselines: As requested by Reviewer V9Me, we added two baselines, which underperform GiFlow as shown in the response.
- Discussion on consistency model: As asked by Reviewer DP7y, we added a discussion on the consistency model.
- Other issues: The other concerns are mainly about the clarification of a specific part of the paper. For example, Reviewer ypzX asks whether there are other flow-based methods, while Reviewer DP7y asks for more illustration on Figure 1. We have clearly clarified these things and revised the manuscript accordingly.

Remaining Unresolved Issues:
- **Reviewer CoSZ**: The reviewer has not yet engaged in any discussion. The main concerns are: (1) motivation for using diffusion or flow matching models for spatiotemporal imputation; (2) GiFlow's applicability to uncertainty estimation. In our response, we have clarified the motivation and emphasized that although uncertainty estimation is important in some applications, it is an orthogonal direction to the focus of this paper.
- **Reviewer ypzX**: The reviewer confirms that our first response ``solves most of the concerns" (Reviewer ypzX). The only remaining question is whether the performance improvement comes from higher parameter numbers. We answered this by providing a new experiment demonstrating that GiFlow has fewer parameters compared to other diffusion-based approaches and is significantly faster in inference, which clearly addresses the reviewer's concern.
- **Reviewer DP7y**: The reviewer has no remaining questions and increased the score to 8.
- **Reviewer V9Me**: After the first response, the remaining concerns are (1) the performance improvement is not significant enough; (2) the model may not be scalable. Regarding these critiques, we would like to stress that (1) As acknowledged by other reviewers, GiFlow already presents strong performance across extensive experimental settings. Expecting a new method to achieve state-of-the-art performance across all settings, and have significant performance gain, while simultaneously being more efficient is not a realistic criterion. (2) The efficiency of GiFlow is demonstrated both theoretically and empirically, but we would like to clarify that efficiency does not necessarily imply scalability, and scalability is not a claimed contribution of our work.

Summary: We conducted all suggested experiments and incorporated them, along with detailed clarifications addressing the reviewers’ questions, into the revised manuscript. After these changes, Reviewers ypzX and DP7y acknowledged that we have addressed their initial concerns. Reviewer DP7y increased the score, leading to (8, 4, 4, 2). Reviewer ypzX raised a follow-up question, which we clearly addressed right after it.  During the discussion period, we didn't receive more responses from the three reviewers who kept the score. We believe our responses have addressed all the concerns. We appreciate all reviewers for their valuable feedback, and we again thank the AC for your time and effort.

---

### Meta-Review · Area_Chair_U2Hy · 2026-01-02

**Summary:**

In summary, there are concerns from the reviewers, regarding the incremental nature of the contribution, the lack of significant performance gains, and the absence of uncertainty quantification.

This paper successfully integrates Graph-Informed priors into a Flow Matching (FM) and proposes a Graph-Informed Flow Matching framework for spatiotemporal imputation. Reviewer CoSZ pointed out that this represents a natural and straightforward extension of flow matching’s flexibility rather than a fundamentally new method. Reviewer V9Me emphasized that $t$-test results do not demonstrate broad statistical significance across most datasets and metrics. Besides, the authors ignore uncertainty estimation, which limits the model's utility compared to the diffusion-based methods.

**Reviewer Concerns:**

The authors provided runtime and parameter analyses (Reviewer ypzX and DP7y); they included experiments on block-missing patterns in PeMS data and added recent baselines (ImputeFormer, CoFILL); the authors provided an extra validation of the transport cost reduction (Theorem 1).

Reviewer CoSZ’s comments regarding the lack of uncertainty estimation shows it is a significant limitation for a generative model in this domain. Reviewer V9Me’s concern regarding marginal performance gains remains unresolved; the authors argue that state-of-the-art performance is not a "realistic criterion" when coupled with efficiency. The statistical significance is narrow. Also, the scalability of the $O(N^2)$ complexity to larger spatiotemporal graphs still remains a bottleneck.

**Reviewer Scores:**

CoSZ 4

DP7y 8

ypzX 4

V9Me 2

---

### Decision · Program_Chairs · 2026-01-26

Reject